

# Spatio-temporal soil moisture retrieval at the catchment-scale using a dense network of cosmic-ray neutron sensors

Maik Heistermann[1], Till Francke[1], Martin Schrön[2], and Sascha E. Oswald[1]

[1]Institute of Environmental Science and Geography, University of Potsdam, Karl-Liebknecht-Straße 24–25, 14476 Potsdam, Germany
[2]UFZ – Helmholtz Centre for Environmental Research GmbH, Dep. Monitoring and Exploration Technologies, Permoserstr. 15, 04318, Leipzig, Germany

**Correspondence:** Maik Heistermann (maik.heistermann@uni-potsdam.de)

**Abstract.**

The method of Cosmic-Ray Neutron Sensing (CRNS) is a powerful technique to retrieve representative estimates of soil water content at a horizontal scale of hectometers (the "field scale") and depths of tens of centimeters ("the root zone"). This study demonstrates the potential of the CRNS technique to obtain spatio-temporal patterns of soil moisture beyond the integrated volume from isolated CRNS footprints. We use data from an observational campaign between May and July 2019 which featured a network of more than 20 neutron detectors with partly overlapping footprints in an area that exhibits pronounced soil moisture gradients within $1\,\mathrm{km}^2$. The present study is the first to combine these observations in order to represent the heterogeneity of soil water content at the sub-footprint scale as well as between the CRNS stations. First, we apply a state-of-the-art procedure to correct the observed neutron count rates for static effects such as sensor sensitivity, vegetation biomass, soil organic carbon and lattice water, as well as for the influence of the temporally dynamic factors barometric pressure, air humidity, and incoming flux of neutrons. Based on the homogenised neutron data, we investigate the robustness of a uniform calibration approach using one calibration parameter value across all CRNS stations. Finally, we benchmark two different interpolation techniques in order to obtain space-time representations of soil moisture: first, Ordinary Kriging with a fixed range; second, a heuristic approach that complements the concept of spatial interpolation by the idea of a geophysical inversion ("constrained interpolation"). For the latter, we define a geostatistical model of the spatial soil moisture variation in the study area, and then optimize the parameters of that model so that the error of the forward-simulated neutron count rates is minimized. In order to make the optimization problem computationally feasible, we use a "heuristic" forward operator that is based on the physics of horizontal sensitivity of the neutron detector. The comparison with independent measurements from a cluster of soil moisture sensors (SoilNet) shows that the constrained interpolation approach outperforms Ordinary Kriging by putting a stronger emphasis on horizontal soil moisture gradients at the hectometer scale. The study demonstrates how a CRNS network can be used to generate consistent interpolated soil moisture patterns that could be used to validate hydrological models or remote sensing products.





# 1 Introduction

## 1.1 The retrieval of soil water content from cosmic ray neutrons

The observation of soil water content remains a scientific challenge. Many methods allow for the pointwise measurement of soil water content, but their spatial representativeness is limited when the small spatial measurement support (in the order of centimeters) is confronted with a high small-scale variability of soil moisture. Remote sensing, in turn, can provide area-integrated measurements, but is typically limited by shallow penetration depths, low overpass frequencies, and the interference of vegetation and surface roughness, to name only a few.

Over the past decade, Cosmic-Ray Neutron Sensing (CRNS) has been established as a powerful alternative to retrieve volume-integrated estimates of soil water content (Zreda et al., 2008; Desilets et al., 2010; Zreda et al., 2012). These estimates are considered as representative for a footprint that extends horizontally over several hectometers (the "field scale"), and vertically over several decimeters ("the root zone"), at a temporal resolution of 3–12 hours (Schrön et al., 2018), depending on detector sensitivity and altitude, for instance. The method relies on measurements of the ambient density of "epithermal"

neutrons (at energies of $1–10^5$ eV) above the ground which is inversely related to the presence of hydrogen and hence soil moisture (Köhli et al., 2020).

The soil water content is mostly inferred from the intensity of epithermal neutrons by using a transfer function such as the one suggested by Desilets et al. (2010), and requires the calibration of a parameter $N_0$ on independent measurements of soil water content in the footprint of a neutron detector (see Schrön et al., 2017, for a recent synthesis). The value of $N_0$ is affected by a

variety of factors, including topography, spatial heterogeneity of soil water content, and the sensitivity of the detector (Fersch et al., 2020a; Schrön et al., 2018), but also the occurrence of hydrogen in snow (Schattan et al., 2017), vegetation (Baroni and Oswald, 2015), lattice water, litter (Bogena et al., 2013), or soil organic carbon.

## 1.2 Beyond soil moisture retrieval in isolated sensor footprints

Until today, CRNS studies have focused on "isolated" sensor footprints only. As an extension to that approach, CRNS roving

has demonstrated potential to detect patterns of soil water content along transects across the landscape (Schrön et al., 2018, see, e.g.,). For that purpose, the sensor is slowly moved through the landscape by using a suitable vehicle. Accordingly, CRNS roving has been successful in revealing spatial patterns beyond the plot scale (Franz et al., 2015; McJannet et al., 2017). Yet, roving can just produce snapshots in time. Furthermore, the choice of transects is typically limited by the network of accessible pathways - which could reduce the representativeness of soil moisture estimates especially due to the influence of the road

material (Schrön et al., 2018).

The present study explores another approach to continuously monitor the spatial distribution of the soil water content at a horizontal scale of hectometers. The main idea is to cover an area of interest with a dense network of stationary CRNS sensors. The term "dense" suggests that the footprints of the CRNS sensors should ideally overlap, or at least adjoin each other, which implies that the distances between the sensors should be in the order of one footprint radius. Based on the observations of

such a dense network, we aim to investigate spatial patterns of soil water content, and to explore different ways to translate





soil moisture estimates from individual footprints into consistent space-time representations of soil moisture across a study domain.

In order to allow for such an investigation, a "Joint Field Campaign" (JFC) was carried out from May to July 2019 by the eight member institutions of the Cosmic Sense project consortium. The JFC featured more than 20 neutron detectors in the
Upper Rott catchment - an area of just 1 km$^2$ that is characterised by heterogeneous land use and strong soil moisture gradients. In the course of the campaign, the soil water content was highly dynamic: the soils were saturated after an exceptionally wet May, followed by a general period of drying in June and July, that was interrupted, from time to time, by short events of heavy rainfall.

Recently, all data collected in this JFC have been made available to the public (Fersch et al., 2020a). The present study is
the first to explore this unique data set regarding its potential to retrieve space-time representations of soil water content at the hectometer scale. More specifically, we aim at addressing the following issues:

1. Can we find a consistent calibration parameter $N_0$ for all CRNS stations? The study area is characterised by a strong spatial heterogeneity e.g. with regard to vegetation (forest vs. meadows), soil types (peat soils vs. loam), terrain (hill slopes vs. valley bottom), and the existence of below-ground structures with low permeability. Consequently, soil mois-
ture as well as epithermal neutron count rates exhibit a strong variation in space. Furthermore, CRNS sensors from different manufacturers, and with different detection techniques and sensitivities were deployed in the campaign. Hence, the first question of this study is whether we can - in the face of this strong heterogeneity - apply a uniform approach to convert neutron intensity to soil water content; or, more specifically, whether we can find, for the key parameter of that conversion, $N_0$, a uniform value that allows to consistently represent the observed soil moisture. This goal would require
to homogenise the observed neutron count rates by accounting for various factors, some of which are rather dynamic in space (such as vegetation biomass, soil organic carbon, sensor sensitivity) while others are more dynamic in time (such as barometric pressure, humidity, or incoming neutron fluxes). Furthermore, we examine how the uncertainty of these factors affects the uncertainty of our $N_0$ estimation. It is important to better understand these uncertainties before comparing CRNS-derived soil moisture dynamics at different locations.

2. What do the differences between the soil moisture estimates at the CRNS footprint scale tell us about soil moisture patterns at the catchment scale? How temporally persistent are these spatial patterns, and are they consistent with our physical expectations? We address these questions based on the uniform soil moisture estimation procedure, including a uniform (or "joint") $N_0$ and explore the spatio-temporal variation of soil moisture as observed by the CRNS sensors, i.e. the variation of soil moisture between the sensor footprints.

3. How does soil moisture vary within and between the sensor footprints? To address this question, we examine two differ- ent interpolation techniques: as a benchmark, the CRNS-derived soil moisture estimates are interpolated using Ordinary Kriging with a fixed range. Alternatively, we introduce a technique that complements the concept of spatial interpola- tion by the idea of a geophysical inversion. To that effect, we define a geostatistical model of the spatial soil moisture variation in the study area, and then optimize the parameters of that model so that the difference between observed and





the forward-simulated neutron count rates is minimized. In order to make the optimization problem computationally feasible, we use a "heuristic" forward operator that is based on the horizontal sensitivity pattern of the CRNS sensor. Essentially, this implies to generate additional spatial information by using nothing more but plausible assumptions. In order to verify the potential of such an approach, we evaluate the resulting soil moisture maps by independent measurements from a cluster of conventional soil moisture measurements by FDR (SoilNet) in the north-western part of the study

area.

The manuscript is structured as follows: Fersch et al. (2020a) have already provided a comprehensive and detailed scientific and technical description of the study site and the data set collected during the JFC. That is why we will just provide a very brief summary of the Upper Rott catchment (section 2), and the data that was actually used in the present study (section 3). In section 4, we then outline the various processing steps required to address the above research questions: the homogenisation

and correction of neutron counts; the calibration of $N_0$ based on data collected in a manual sampling campaign, and the corresponding estimation of soil moisture time series for each CRNS sensor; and the interpolation of CRNS-based soil moisture at the catchment scale. We then present the corresponding results (section 5) and conclusions (section 6).

## 2 Study site

The Fendt study site, at an altitude of around 595 m ASL, is located in the headwater of the Rott river (Fig. 1). The catchment

has an area of about 1 km$^2$, and is part of the TERENO Pre-Alpine Observatory (Kiese et al., 2018).

Gravels, together with fractions of loam and silt, dominate at the slopes of the valley. Towards the draining rivulet at the lower elevations, loamy and silty sediments can be found, as well as some occurrences of peat (Fersch et al., 2018). The hydraulic heads of the shallow aquifers range from 4 to 0.2 m below the ground. Cambic Stagnosol is the dominant soil class, and the fractions of sand, silt and clay comprise 27, 41, and 32 % (Kiese et al., 2018). Grassland is the most important land use,

with particularly wet spots along the rivulets in the north and the south. A mixed, heterogeneously structured forest extends mainly along the eastern slopes of the catchment.

## 3 Data

### 3.1 Overview

This section gives a brief summary of the data that has actually been used in the present analysis. As already pointed out,

Fersch et al. (2020a) have provided a comprehensive description that should serve as a detailed reference.

### 3.2 CRNS measurements

In total, 24 stationary CRNS detectors were positioned in the study area from May to July 2019. For various organisational and technical reasons, some sensors recorded only briefly or patchily. For the present analysis, we only use those 18 sensors which



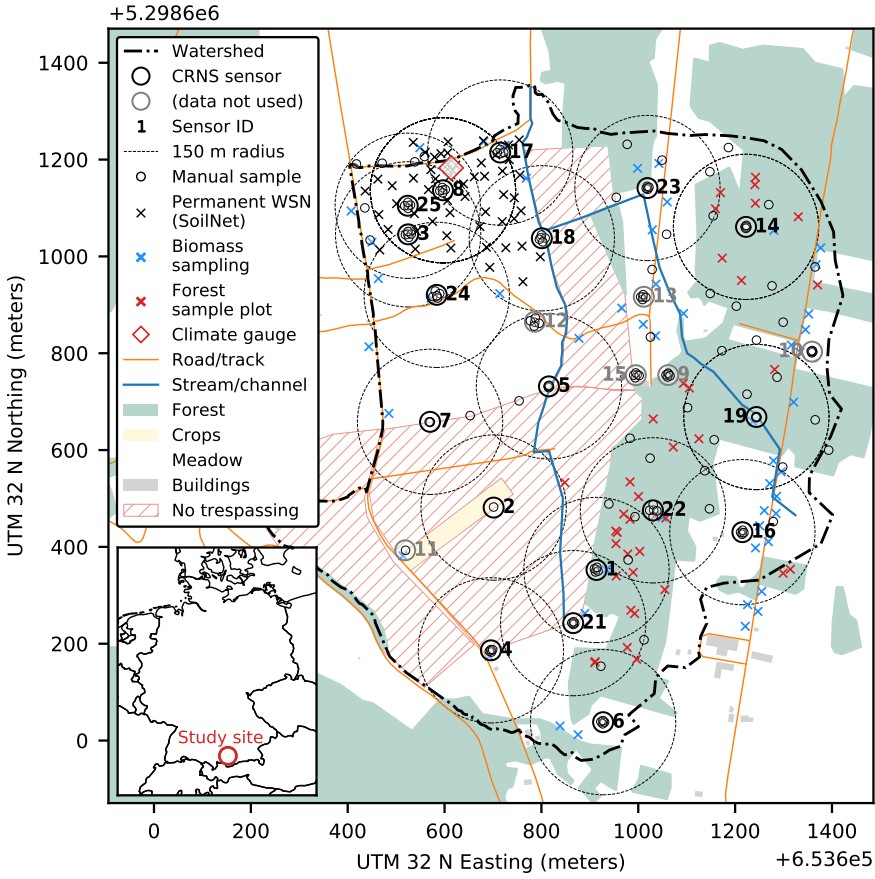

**Figure 1.** Study area around Fendt with the headwater catchment of Rott river. With regard to land cover, the forest, a few settlements, and a cropland plot are highlighted – the rest is mostly grassland. For instrumentation, we show the CRNS-locations and a typical (Schrön et al., 2017) footprint radius of 150 m, the climate station, and the locations of soil moisture measurements from the SoilNet and from the manual measurement campaign from June 25 to 26, 2019. See section 3.7 on the usage of OSM basemap data (© OpenStreetMap contributors, 2020, distributed under ODbL license).

coherently covered at least the majority of the study period including the manual sampling campaign at the end of June 2019.

Tab. 1 gives an overview of those sensors, and Fig. 1 shows their locations. 17 sensors were manufactured by Hydroinnova LLC: models CRS-1000 and CRS-2000 use $^3$He as a detector gas, while models CRS-1000-B and CRS-2000-B are based on $^{10}$BF$_3$. Another sensor was manufactured by Lab-C LLC, and uses, as a neutron converter, a multi-wire proportional chamber with solid $^6$Li. For all these detectors, the "thermalization" (i.e. slowing down epithermal neutrons to detect them) is facilitated by a moderator material around the detection chamber (see e.g. Zreda et al., 2012, for details).

The placement of the CRNS sensors had been based on a set of constraints, some of which scientific, some legal or practical, and some also conflicting with each other (Fersch et al., 2020a): e.g., we aimed to maximise the spatial coverage of the





**Table 1.** Overview of the 18 CRNS sensors used in this study, incl. manufacturer, sensor model, and underlying technology; the main land use in the sensor footprint; and the sensitivity of the sensor relative to the calibrator probe (#20). Please note that the sensor IDs in this table are not consecutively numbered for consistency with Fersch et al. (2020a).

| ID | Manufacturer | Sensor model | Technology | Dominant land use | Sensitivity |
|----|----|----|----|----|----|
| 1 | Hydroinnova | CRS 2000-B | $^{10}BF_3$ | forest, meadow | 1.190 |
| 2 | Hydroinnova | CRS 1000 | $^{3}He$ | crops, meadow | 0.452 |
| 3 | Hydroinnova | CRS 1000 | $^{3}He$ | meadow | 0.458 |
| 4 | Lab-C | NeuSens dual | $^{6}Li$ | meadow | 4.530 |
| 5 | Hydroinnova | CRS 1000-B | $^{10}BF_3$ | meadow | 0.670 |
| 6 | Hydroinnova | CRS 1000-B | $^{10}BF_3$ | meadow, forest | 0.668[*] |
| 7 | Hydroinnova | CRS 1000-B | $^{10}BF_3$ | meadow | 0.668[*] |
| 8 | Hydroinnova | CRS 2000-B | $^{10}BF_3$ | meadow | 1.161 |
| 14 | Hydroinnova | CRS 2000 | $^{3}He$ | forest | 0.871 |
| 16 | Hydroinnova | CRS 2000-B | $^{10}BF_3$ | meadow | 1.148 |
| 17 | Hydroinnova | CRS 2000-B | $^{10}BF_3$ | meadow | 1.121 |
| 18 | Hydroinnova | CRS 1000 | $^{3}He$ | meadow | 0.414 |
| 19 | Hydroinnova | CRS 2000-B | $^{10}BF_3$ | forest | 1.147[*] |
| 20 | Hydroinnova | Calibrator | $^{3}He$ | – | 1.000 |
| 21 | Hydroinnova | CRS 2000-B | $^{10}BF_3$ | meadow, forest | 1.132 |
| 22 | Hydroinnova | CRS 2000-B | $^{10}BF_3$ | forest | 1.168 |
| 23 | Hydroinnova | CRS 2000-B | $^{10}BF_3$ | meadow, forest | 1.127 |
| 24 | Hydroinnova | CRS 2000-B | $^{10}BF_3$ | meadow | 1.138 |
| 25 | Hydroinnova | CRS 1000-B | $^{10}BF_3$ | meadow | 0.665 |

[*] Calibrator not available, so the sensitivity was obtained from the average values of the same models.

catchment, but also the overlap between sensor footprints. As a compromise, the CRNS sensors were placed more densely in the north-western part of the study area where the SoilNet observations could be utilized to validate spatial patterns of soil water content (see Fig. 1).

The sensitivity of the detectors used in this study ranged over an order of magnitude (between the most sensitive one, NeuSense Dual, and the least sensitive, CRS-1000). However, the effective sensitivity could vary between sensors of the same type, too. In order to standardize neutron count rates across sensors, we collocated a mobile "calibrator" sensor (#20 in Tab. 1) consecutively beside most of the stationary probes for a duration of at least one day.

    As another sensitivity reference, we used mobile CRNS measurements carried out from June 25 to 26. In these two days, a
roving CRNS (see Fersch et al., 2020a, for details) was placed next to the stationary CRNS sensors (except sensors 1, 21, and 22 due to inaccessibility) for a duration of at least 30 minutes. The high sensitivity of the roving CRNS allowed count rates to be sufficiently representative to relate them to the count rates of the stationary sensors.





## 3.3 Incoming cosmic-ray neutron flux

In order to account for the variation of the incoming cosmic-ray neutron flux, Schrön et al. (2016) recommend to select
recordings from neutron monitors which have a similar cutoff rigidity as the study location. In this study, we used the monitor
at Jungfraujoch in order to quantify the reference flux. The data was obtained from the Neutron Monitor Database (monitor ID
"JUNG", http://www.nmdb.eu).

## 3.4 Local measurements of soil water content and other soil variables

As mentioned in section 1.1, we require local measurements of the soil water content and the soil bulk density in order to
calibrate the relationship between epithermal neutron intensity and volumetric soil water content. The same holds for the
verification of soil moisture retrievals from observed neutron intensities. During the JFC, several measurement techniques
were applied to meet these requirements.

### 3.4.1 Permanent soil sensor network (SoilNet)

The north-west of the catchment is permanently equipped with a SoilNet (version 3, SoilNet (2018)), a network of 55 vertical
sensor profiles (locations shown in Fig. 1). Temperature and permittivity are recorded in depths of 5, 20, and 50 cm, at an
interval of 15 minutes, redundantly at each depth with two slightly displaced sensors. Please see Fersch et al. (2020a) for
further details on how volumetric soil moisture was obtained from permittivity.

### 3.4.2 Manual soil sampling and measurement of soil water content

Vertical measurements were carried out from June 25 to 26, 2019, with depth increments of 5 cm down to a maximum depth
of 30 cm. Right beside each stationary CRNS sensor, a vertical profile was measured using soil cores and thermo-gravimetry.
At another 139 locations at and in-between the stationary CRNS sensors, vertical profiles using the same depths were obtained
from manual FDR measurements. Fig. 1 provides an overview of the sampling locations. Please see Fersch et al. (2020a) for
a comprehensive description of the sampling scheme, the sample processing for thermo-gravimetry, and the conversion of
measured permittivity to volumetric water content for the FDR measurements.

## 3.5 Vegetation and biomass

For grassland and cropland, above-ground biomass was measured at three dates (May 14–16, June 6 and July 17) at the same
45 locations, and the water content was determined by drying to a constant weight at 65 °C.

   For quantifying above-ground biomass in the forest, a different approach was used. Ground-truth information from 116 sites
with representative species composition allowed the derivation of a tree-species map using multitemporal RapidEye-imagery.
In addition, detailed *tree-based* surveys at 16 plots (incl. species, height and breast-height-diameter, see Fersch et al. (2020a)
for details) yielded values of above-ground biomass estimates. Using stand-height information from LIDAR, these estimates
were generalized for the entire forested area (Stockmann, 2020).





### 3.6 Meteorological data

The location of the meteorological station at the Fendt site is shown in Fig. 1. The observed meteorological variables required
for the present analysis include precipitation, barometric pressure, as well as air relative humidity and temperature, recorded at
a temporal resolution of one minute.

### 3.7 Land use, roads, waterways

For visualisation, OSM layers (OpenStreetMap contributors, 2020) from http://download.geofabrik.de were used, incl. water-
ways, landuse, and roads (distributed under ODbL license, www.openstreetmap.org/copyright).

## 4 Methods

### 4.1 Homogenisation of neutron intensities in space and time

In order to make the observed neutron count rates comparable across space and time, different effects need to be taken into
account, namely the different detector sensitivities, atmospheric effects, and the effects of hydrogen pools other than soil water.

#### 4.1.1 Standardization of sensitivity

Measurements of a collocated calibrator probe (see 3.2) served to standardize CRNS measurements. For the period of collo-
cation, the ratio between the corresponding stationary and mobile neutron counts was defined as the "sensitivity factor" which
was then applied to standardize – to the calibrator level – the neutron intensities recorded by a stationary detector. In case a
calibrator collocation was missing, average values of the same sensor model type were used instead (see Tab. 1).

#### 4.1.2 Accounting for atmospheric effects

The observed neutron count rates are affected by a range of dynamic atmospheric variables that need to be accounted for in order
to make neutron intensities comparable across time. These dynamic variables include atmospheric vapor content, barometric
pressure, and incoming cosmic-ray neutron flux. To correct for these effects, we used the data mentioned in sections 3.6 and
3.3. The corresponding standard correction approach is outlined e.g. by Scheiffele et al. (2020) in Appendix A.

#### 4.1.3 Accounting for the effects of vegetation

In order to allow for the comparison of neutron intensities across locations, we accounted for the effect of above-ground
biomass and its variability in space. We assume that this variability is dominated by the presence of woodland in contrast to
grassland. In fact, the average above-ground dry matter for grassland and cropland amounted to $0.2 \, \text{kg/m}^2$ while the average
above-ground dry mass for forested area was quantified as $24.4 \, \text{kg/m}^2$ (Stockmann, 2020). The spatial variability of forest
biomass turned out to be high. However, its quantification at high spatial resolution (i.e., order of $100 \, \text{m}^2$) included consid-





erable uncertainties. Therefore, we decided to use only the *average* above-ground biomass estimates for forest and grassland,
respectively. The corresponding spatial distribution of above-ground biomass (in kg/m$^2$), based on vegetation cover being
either forest or grassland, was represented on a regular grid with a horizontal resolution of 10 m.

Based on this gridded data, we computed the weighted average of dry vegetation matter per CRNS sensor footprint by using
the horizontal weighting function $W_r/r$ as suggested by Schrön et al. (2017). We assumed the resulting average dry matter per

sensor footprint to be constant throughout the duration of the campaign. Based on that estimate, we accounted for the effect of
vegetation on neutron count rates by following Baatz et al. (2015) who suggested "a neutron intensity reduction of 0.9 % per
kg of [dry] biomass per m$^2$".

### 4.1.4   Estimating average soil properties per sensor footprint

As pointed out in 3.4.2, the vertical distribution of volumetric water content $\theta$, soil bulk density $\rho_b$, soil organic matter content

$SOM$, and lattice water content $LW$ was determined at each CRNS sensor from thermo-gravimetry. For volumetric water
content, additional vertical profiles were measured using the FDR technique. For the estimation of $N_0$ (see section 4.2), rep-
resentative averages of these variables have to be obtained for each CRNS footprint. For that purpose, we first approximated
the 3-dimensional distribution of soil properties from the available measurements, and then applied the vertical and horizontal
weighting functions provided by Schrön et al. (2017) in order to compute weighted averages. To this end, the following steps

were carried out:

**Fit vertical profiles**

In order to generalize the vertical distribution of soil variables $v_i$ (i.e. $\theta$, $\rho_b$, $SOM$, $LW$) across the study area, we fitted a
piecewise linear function to each profile and variable: from the soil surface (0 cm) down to a depth of 13 cm, the function
assumes a linear change of any variable value from $v_i(0cm)$ to $v_i(13cm)$. Below 13 cm depth, the variable is assumed to

remain constant at a value $v_i(13cm)$. This approach has been found to reflect the typical vertical distribution pattern for all
soil variables fairly well, while reducing spurious effects of outliers when the variables are horizontally interpolated (see next
section). Example profiles are illustrated in the Supplement (Figs. S1-S3).

**Horizontally interpolate the vertical distribution parameters**

The fitted parameters $v_i(0cm)$ and $v_i(13cm)$ at each profile location were then interpolated in space using Ordinary Kriging

with an exponential variogram model and a range (i.e. a spatial autocorrelation length) of 50 m (150 m for soil moisture) on a
10 x 10 m grid. Based on the vertical distribution function and the interpolated parameters, we could then compute the value
of the soil variable at 1 cm vertical resolution between 0 and 30 cm. It should be noted that in this study, we applied Ordinary
Kriging rather "heuristically" than in a typical geostatistical way: the underlying assumption of stationarity as well as the
choice of an exponential variogram model with a specific range was based on the aim to create continuous and plausible spatial

patterns that are robust in areas of extrapolation and in which spatial autocorrelation is explicitly considered (instead of implicit





assumptions used in techniques such as nearest neighbour, inverse distance weighting, (thin-plate) splines or other available methods). The sensitivity of our results to the range parameter will be explicitly investigated in a Monte-Carlo-analysis (see section 4.3).

For soil organic matter and lattice water, the procedure was modified since both variables had not been determined for each
profile separately, but from mixing samples within areas classified as "forest on mineral soil", "other land use on mineral soil" as well as "other land use on organic soil" (see Fersch et al. (2020a)). Hence, the vertical distribution function was only determined for these three classes, and, instead of interpolation, the same vertical profile was assigned to each grid cell based on its membership in one of the three classes.

**Compute the weighted average variable value for each sensor footprint**

In the final step, we used the vertical and horizontal weighting functions, $W_d$ and $W_r$, as presented in Eq. A1 of Schrön et al. (2017), in order to compute average values of the soil variables per sensor footprint. The vertical average was obtained for each grid cell in the footprint based on the vertical weighting function $W_d$ and the horizontal distance $r$ of the grid cell to the sensor. Then, the vertical averages were averaged horizontally based on the horizontal weighting function $W_r$. As the soil variables of interest are parameters of the weighting functions themselves, the weighting procedure was iterated until the average variables
converged (typically after less than 5 iterations), as recommended by Schrön et al. (2017).

### 4.2 Calibration of $N_0$

The soil water content is estimated from the epithermal neutron count rate using the standard transfer function suggested by Desilets et al. (2010). This requires the calibration of a parameter $N_0$ on local soil moisture observations.

$$\theta_g^{all} = \theta_g + \theta_g^{som} + \theta_g^{lw} = \frac{a_0}{N/N_0 - a_1} - a_2, \quad \theta = \frac{\varrho_b}{\varrho_w} \cdot \theta_g \tag{1}$$

In this Eq. 1, subscript $g$ indicates gravimetric soil water (equivalents) in units of kg/kg; hence, $\theta_g$ is the gravimetric soil water content, $\theta_g^{som}$ and $\theta_g^{lw}$ are the gravimetric soil water equivalents of soil organic matter and lattice water, $\theta$ is the volumetric soil water content (in m³/m³), $\rho_w$ and $\rho_b$ are the density of water (assumed as 1000 kg/m³) and the soil bulk density (in kg/m³). $N$ is the corrected neutron intensity (in counts per hour, cph), $N_0$ (in cph) is the calibration parameter. The shape parameters $a_0$, $a_1$, and $a_2$ could be adapted to specific local conditions, but they have also proven to be robust in many previous studies
(Desilets et al., 2010; Evans et al., 2016; Schrön et al., 2017, among others), and where set to $a_0 = 0.0808$, $a_1 = 0.372$, and $a_2 = 0.115$. Please note that water equivalents from vegetation ($\theta_{bio}$) are not included in the sum on the left since the effects of vegetation biomass were already accounted for in the corrected neutron intensity $N$ (see section 4.1.3).

In order to use Eq. 1 to compute $\theta_g(N)$ or $\theta(N)$, respectively, from observed neutron count rates $N$, $\theta^{som}$ and $\theta^{lw}$ as well as the soil bulk density $\varrho_b$ need to be independently quantified as weighted averages for each CRNS sensor footprint (see
section 4.1.4 for details). For any calibration of $N_0$, $\theta$ needs to be independently quantified as calibration reference which we





refer to as $\theta^{obs}$. $\theta_i^{obs}$ was computed as a weighted average for each CRNS sensor footprint $i$ from manual measurements (also outlined in section 4.1.4).

Usually, $N_0$ is calibrated for a single CRNS sensor at a particular site. For our dense cluster, however, we calibrated Eq. (1) by assuming a spatially uniform value of $N_0$ across the study area. Hence, we optimized $N_0$ by minimizing the mean absolute
difference between the average volumetric soil moisture $\theta_i^{obs}$, and the corresponding $\theta(N_i, N_0)$, that can be obtained by solving Eq. 1 for $\theta$ and using the corrected neutron intensities $N_i$:

$$\underset{N_0}{\arg\min} \sum_{i=1}^{18} \left| \theta_i^{obs} - \theta(N_i, N_0) \right| \tag{2}$$

The manual soil moisture measurements had been carried out over two consecutive days, June 25-26, 2019 (see section 3.4). Hence, the temporally varying parameters (neutron count rates $N_i$, barometric pressure, humidity) required as inputs to the
weighting functions and Eq. 2 were obtained by computing temporal means between June 25, 08:00 (UTC) and June 26, 18:00 (UTC).

### 4.3 Exploring the uncertainty of $N_0$ calibration

Obviously, uncertainties from various sources affect our estimate of $N_0$. These uncertainties arise from errors in the measurements of soil moisture and neutron fluxes as well as the validity and parameterisation of functional relationships and underlying
assumptions. In order to get, on the one hand, a better idea about the robustness of our $N_0$ estimate, and, on the other hand, about the local uncertainty of observed and CRNS-based soil moisture, we carried out a Monte-Carlo analysis (see also Jakobi et al., 2020). In that analysis, we repeated the $N_0$ calibration (see Eq. 2) for 200 times, each time with a set of randomly disturbed input parameters. These parameters and disturbances (i.e. assumed error distributions) are as follows:

- *Sensor sensitivity*: As pointed out in section 4.1.1, the neutron count rates of all sensors were standardized to the level of
a "calibrator" sensor. Yet we found some variability of the resulting sensitivity factors even for the same sensor type. In section 5.2, we will also evaluate the consistency of the standardization by using independent roving measurements. In the Monte-Carlo-analysis, we assume the sensitivity to vary by $\pm 2\%$ with respect to the sensitivity level shown in Tab. 1.

- *Averaging period for $N_i$*: The calibration of $N_0$ requires, for each CRNS sensor $i$, an average neutron intensity $N_i$ for
the corresponding calibration period. The definition of the time window over which to compute that average is always subject to some arbitrariness. In the present case, it is even more difficult, as the manual soil moisture measurements took place at some time within a period of two consecutive days. In the previous section, we suggested to average between June 25, 08:00 (UTC) and June 26, 18:00 (UTC). In order to capture the effect of other time windows, we randomly selected 12-hour-windows within this time span, and computed the average $N_i$ from these windows.

- *Dry vegetation matter*: While the grassland biomass can also vary in space and time, the spatial variability of woodland biomass is much more uncertain (and relevant). Stockmann (2020) has attempted to represent the spatial variability by





combing allometric approaches and remote sensing, and found the dry matter mass to vary between 13 - 73 $kg/m^2$, with a relative error of 17 % at the plot scale. As we had assumed a homogeneous distribution of forest biomass (see section 4.1.3), we have to expect large local errors. In our Monte-Carlo-analysis, we will vary the dry matter mass per CRNS footprint within $\pm 20\%$ with respect to the estimated value.


 – *Water equivalent*: As pointed out in section 4.1.4, soil organic matter and lattice water content were only determined from mixed samples for three different land-use/soil combinations. Hence, the local values for the resulting water equivalents could substantially depart from these average values, so that we assume the estimated values of $\theta_{som}$ and $\theta_{lw}$ per CRNS footprint to vary within $\pm 20\%$ of the initial estimate.


 – *Kriging range*: as pointed out in section 4.1.4, the choice of using an exponential variogram model together with a specific Kriging range for specific variables was rather pragmatic. In order to investigate potential effects, we varied the Kriging range for soil moisture between 50 and 500 m, and for bulk density between 30 and 70 m.

 – *Subsampling from soil profiles*: For the $N_0$ calibration, the vertical soil moisture distribution had been sampled at 160 locations in the study area. Yet, given the size of the area and the number of CRNS sensors, the corresponding sample size per CRNS sensor is still substantially lower as compared to previous studies (18 samples per footprint were recommended


by Zreda et al. (2012)). In order to analyse how the availability (or, inversely, the absence) of sampling locations, we employed a sub-sampling approach: for each realisation in our Monte-Carlo analysis, we used only 80 % of our sampling locations for the interpolation of bulk density and volumetric soil moisture. These 80 % were randomly selected from the entire population of locations.

**4.4 Soil moisture retrieval and spatial interpolation**

To improve the signal-to-noise ratio for each CRNS sensor, we computed the average neutron count rates at an interval of 24 hours, and then converted, for each CRNS sensor, this averaged neutron count rate to daily volumetric soil water content by using Eq. 1 with the calibrated $N_0$. This soil water content will be referred to as $\theta(N_i)$ in the following. These values will be examined as a first order representation of how the soil water content varies in the study area in space and time.

In order to represent the spatial soil moisture distribution within the catchment, i.e. inside and in-between CRNS footprints, we interpolate $\theta(N_i)$ to a 10 m x 10 m grid that spans the entire study area. The grid resolution is arbitrarily selected, and does not necessarily reflect the resolution at which the grid effectively conveys information of spatial heterogeneity; in other words, the product should not be interpreted at the scale of 10 m. Still, we require this comparatively fine horizontal resolution since some of the following steps require to re-aggregate (i.e. to average) the spatial soil moisture estimates inside a CRNS footprint.

Gridded soil moisture estimates are obtained by two interpolation approaches (or "models") which we will refer to as "unconstrained" and "constrained" (please note that the following explanations refer to interpolation in space for a given point in time; temporal (auto-)correlation is not considered here).

**Unconstrained model**



What we refer to as the "unconstrained" approach could imply any kind of (geostatistical) model or assumption $m$ that
represents the spatial distribution of soil moisture, $\theta$, on the basis of any parameter set $\mathbf{p}$. For example, $m(\mathbf{p})$ could be the
nearest neighbour algorithm. In that case, $\mathbf{p}$ would be the soil moisture values at a set of sampling points. As another example,
$m(\mathbf{p})$ could be a statistical relationship between landscape attributes and soil moisture, hence $\mathbf{p}$ would comprise the parameters
of that statistical model. Or, $m(\mathbf{p})$ could be a physical model of water movement in soils, with $m(\mathbf{p})$ being the entirety of
(potentially spatially distributed) model parameters.

Altogether, $m$ could be an interpolation algorithm, a statistical or physical model, or any combination of these. What these
approaches have in common is that we initially assume $\mathbf{p}$ (or subsets thereof) to be unknown - although we might be able to
take a plausible guess.

In this study, let us assume that the spatial distribution of soil moisture in the study area is smooth and continuous, and that
this spatial pattern could be represented by a model $m$ that corresponds to Ordinary Kriging with an exponential variogram
model and a range parameter of, say, 300 m, using the CRNS sensor locations as points of support. We hope it is clear to the
reader that the choice of such a model is arbitrary and subjective, although it should be based on our "expert" notion of how
soil moisture varies at a specific scale. If we have a given number of $n$ CRNS sensors in our study area, our resulting model, in
this case, $m(\mathbf{p})$ will have $n$ parameters, too, namely the soil moisture values $\theta_i$ at the CRNS locations $i$. As pointed out above,
we could take a guess at $\mathbf{p}$. An obvious guess would be to use the CRNS-based soil moisture values, $\theta(N_i)$, at the points of
support.

While that approach is a straightforward interpolation, it has one major drawback: it does not account for the consistency
between the obtained spatial soil moisture distribution and the neutron intensities at the sensor locations ($N_i$), i.e. the produced
spatial pattern of soil moisture does not necessarily correspond to the observed count rates. But if we were able to somehow
simulate the neutron intensities that would result from the soil moisture distribution from $m(\mathbf{p})$, we would have a basis to
constrain (or optimize) our parameters $\mathbf{p}$, by maximising the agreement between observed and simulated neutron intensities.
That is the idea behind the "constrained" approach which is outlined as follows.

**Constrained model**

The "constrained" model $m$ is the same as the above "unconstrained", except that we adjust (optimize) our initial guess of $\mathbf{p}$
such that the disagreement (i.e., the sum of absolute differences) between observed neutron count rates, $N_i^{obs}$, and simulated
count rates, $N_i^{sim}$ (obtained from $m(\mathbf{p})$), is minimized (Eq. 3).

$$\arg\min_{\mathbf{p}} \sum_{i=1}^{n} \left| N_i^{obs} - N_i^{sim}(m(\mathbf{p})) \right| \tag{3}$$

Fig. 2 illustrates the idea and effects of a constrained model for a simple one-dimensional interpolation example with two
CRNS sensors and an (unknown) true soil moisture distribution (solid black line, panel 1). This "truth" exhibits a trend over the
spatial domain, but also some local variability. Panel 2 illustrates the sensor locations together with their horizontal sensitivity
pattern (red shadows). Using a suitable suitable forward operator $\mathscr{H}$ (see next section), we can compute the neutron count rates
$N_i^{obs}$ that we would expect the sensors to observe, based on the true soil moisture. These $N_i^{obs}$ constitute the only information



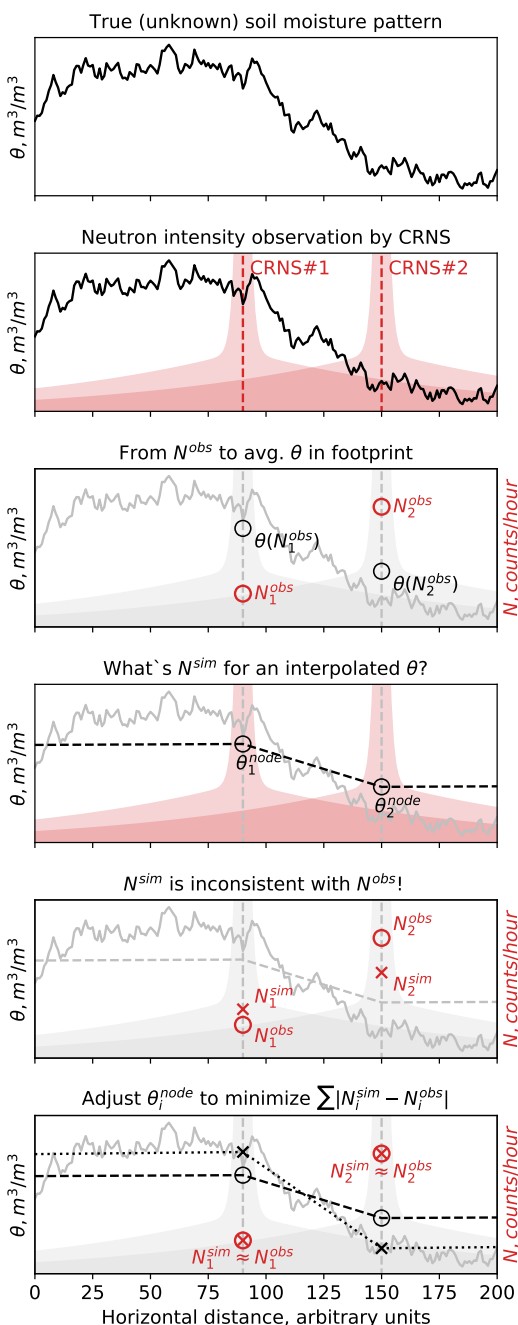

**Figure 2.** Schematic view of the "constrained" interpolation model, illustrated by an idealized example with two CRNS sensors and a 1-D horizontal soil moisture distribution. Symbols are explained in the main text; note that the spatial domain further extends to the right, but is not entirely shown for the sake of clarity.





we have from our CRNS-sensors. From these $N_i^{obs}$, we compute the average soil moisture $\theta(N_i^{obs})$, based on Eq. 1 with a known $N_0$ (panel 3). We then apply the unconstrained interpolation by using $\theta(N_i^{obs})$ as points of support ($\theta_{node,i}$, panel 4). From the resulting soil moisture distribution (panel 4, dashed line), we simulate the corresponding $N_i^{sim}$, again by using the

forward operator $\mathcal{H}$. In the case of a marked soil moisture gradient, as in this example, the resulting $N_i^{sim}$ and observed $N_i^{obs}$ will disagree. The reason for this disagreement is obvious and unsurprising: using the volume-integrated $\theta(N_i^{obs})$ as a *point* of support for the interpolation ($\theta_{node,i}$), we potentially neglect a substantial portion of spatial variation that is hidden behind this volume-average. Hence, spatial gradients will tend to be systematically underrepresented: here, the unconstrained model underestimates areas of high soil moisture in the left half of our spatial domain, and overestimates in the right.

In the "constrained model", we therefore adjust the values for $\theta_{node,i}$ so that the disagreement between $N_i^{sim}$ and measured $N_i^{obs}$ is minimized. This optimisation problem could have an infinite number of equally valid solutions. The problem of equifinality is addressed, however, by using a local optimisation technique (Nelder-Mead Simplex adapted according to Gao and Han (2012)). As our initial parameter guess for ($\theta_{node,i}$) is based on $\theta(N_i^{obs})$, we expect the parameter optimisation to stay close to this initial guess, to change just by as much as required to better represent the local soil moisture gradients. In

section 5, we will examine whether that expectation is met. We also expect that the optimisation problem will be defined better in areas where multiple CRNS sensor have a significant overlap. Conversely, the difference between the initial $N_i^{sim}$ and $N_i^{obs}$ will probably be low for CRNS locations that are rather isolated and distant from others: that is because in the footprint of any isolated (or distant) sensor $i$, the interpolated soil moisture values will be dominated by $\theta(N_i^{obs})$, and less affected by other sensors.

**The forward operator**

In essence, the constrained model resembles to what is generally referred to as a "geophysical inversion", i.e. the identification of parameters $\mathbf{p}$ in a model $m$ by means of inverse simulation: to that end, the observed (integrating) variable (e.g. $N$) is obtained from the estimated target variable (e.g. $\theta$) by means of a physically-based forward operator (simulation). Hence, $\mathbf{p}$ is optimised by minimising the disagreement between the simulation and observation. That way, further spatial information

can be obtained from volume-integrated observations by using our physical understanding of both the observed system and the observation technique itself. On this basis, our "constrained" approach, as outlined above, does not qualify as a "geophysical inversion": first, we do not use a physically-based model to describe our notion of a spatially continuous soil moisture variation at a specific scale. Instead, we use a geostatistical model, namely Ordinary Kriging. Second, we use a rather heuristic implementation of a forward operator $\mathcal{H}$ that should be specified as follows:

$$\mathcal{H} : \Theta_i^m \mapsto N_i^{sim}, \tag{4}$$

where the operator $\mathcal{H}$ simulates the "corrected" epithermal neutron intensity $N_i^{sim}$ at a sensor location $i$ using the entire environment of interpolated (modelled) soil moisture values, $\Theta_i^m$, in a 300 m radius around the sensor, represented as a grid on which each grid cell $j$ has a modelled soil moisture value $\theta_j^m$.





Shuttleworth et al. (2013) developed the COSMIC operator in the context of data assimilation. However, the COSMIC
operator does not account for the horizontal but for the vertical variability of soil moisture, and is therefore not fully eligible
for our purpose. Neutron transport models, in turn, are physically well defined and well justified. Based on Monte Carlo
simulations, they are able to track the histories of millions of neutrons by taking the relevant physical interactions into account.
That requires to consider the heterogeneity of the surrounding environment with regard to all variables that are, in addition
to soil moisture, relevant to the fate of a neutron (e.g. vegetation, topography, atmospheric effects and more). Such neutron
transport models are available, and have proven valid in a wide range of application contexts. Examples are MCNP (Andreasen
et al., 2017), or, more recently, URANOS (Weimar et al., 2020). Both would be perfect forward operators - except that their
computational cost is entirely prohibitive in our application context: the optimisation of $\mathbf{p}$ requires hundreds of iterations which
would require weeks of computation time – for the soil moisture interpolation at one single point in time alone.

Hence, we follow an intermediate approach which we have already referred to as "heuristic": instead of explicitly simulating
the physical interactions of neutrons with the near-surface environment, we use the horizontal weighting function $W_r$ that was
presented in Eq. A1 of Schrön et al. (2017). $W_r$ represents the horizontal sensitivity pattern of a CRNS sensor, and depends
on various environmental, partly dynamic variables such as soil moisture itself, soil bulk density, vegetation, air humidity, and
barometric pressure. Though $W_r$ was not originally intended to support forward simulations, it is based on extensive neutron
transport simulations (Köhli et al., 2015), and therefore has a sufficient physical basis to serve our main purpose: to quantify,
for the observation of neutron intensity at any location $i$, the relative contribution of soil moisture at different distances from the
sensor. $W_r$, however, does not directly yield neutron intensity. From a formal perspective, it is rather a spatial filter function.
In order to obtain a forward operator, we combine $W_r$ with the inverse of Eq. 1, so that we finally get a forward operator as
represented by Eqs. 5 and 6.

$$N_i^{sim} = \mathscr{H}(\Theta_i^m) = N_0 \cdot \frac{a_0}{\langle \Theta_i^m \rangle \cdot \frac{\rho_w}{\rho_{b,i}} + a_2} + a_1 \tag{5}$$

$$\langle \Theta_i^m \rangle = \frac{\sum_j \left( W_{r,i,j} \cdot \theta_j^m \right)}{\sum_j W_{r,i,j}} + \theta_i^{lw} + \theta_i^{som} \tag{6}$$

The usefulness of this heuristic approach is effectively verified in a benchmarking experiment in which we evaluate the
performance of the "unconstrained" versus the "constrained" approach by a comparison with independent observations of the
SoilNet operated in the north-west of the study area.

## 4.5 Comparison to local soil sensor network (SoilNet)

In order to compare the performance of the "unconstrained" and the "constrained" interpolation models, the resulting maps
of average daily soil moisture are compared against daily soil moisture maps obtained from the SMT100 cluster (SoilNet)
operated at the site (Fersch et al., 2020a).





This comparison, however, is not straightforward: While the SoilNet data consist of a set of observations at specific points in space (with a horizontal support of a few centimeters), results of the interpolation models are spatial grids of average soil moisture with varying vertical representativeness.

The SoilNet observations $\theta_{n,z}^{\mathrm{FDR}}$ constitute 55 nodes (index $n$), each of which provides a measurement at three depths $z$ (5, 20, and 50 cm; the two measurements at each depth are averaged). For each node, we first need to obtain a continuous vertical profile: therefore, we linearly interpolate the measurements at intervals of 1 cm between 5 and 20 cm, and between 20 and 50 cm. Between 5 cm and the soil surface, the mean value from 5 cm is used. Second, we have to compute a weighted average $\langle\theta_n^{\mathrm{FDR}}\rangle$ of the vertical profiles obtained at each node, which reflects the soil moisture as detected by the CRNS. In principle, we can use the vertical weighting function $W_d$ from Schrön et al. (2017) for that purpose. However, $W_d$ (or, the penetration depth) is a function of the distance $r$ to a CRNS sensor, too. The interpolation model, in turn, mixes, at any grid point, the soil moisture obtained from various CRNS sensors at different ratios, depending on the distance. Hence, there is no unique definition of $r$ for any SoilNet node. For high soil moisture values around 0.4 m³/m³, however, the penetration depth only decreases by less than 5 cm within a radial distance of 100 m around the sensor (see Fig. 8 in Köhli et al., 2015). Therefore, we decided to simply use the same vertical weights $W_d(r)$ for the entire SoilNet domain, based on a "medium" distance value of $r = 20$ m. Third, we need to organize the vertically averaged soil moisture $\langle\theta_n^{\mathrm{FDR}}\rangle$ in a spatial structure that makes it comparable to the interpolated CRNS-based soil moisture. For that purpose, we horizontally interpolated $\langle\theta_n^{\mathrm{FDR}}\rangle$ to the same grid as the interpolated CRNS-based soil moisture, using the same Ordinary Kriging approach as used for the interpolation of CRNS-based soil moisture (exponential variogram with a range of 300 m).

In order to evaluate the similarity of the spatial soil moisture grids obtained from the FDR-cluster and the interpolation of $\theta(N_i)$, we compute the Spearman's rank correlation between these soil moisture grids for each day from May 20 to July 15, 2019. We use this measure (instead of e.g. the RMSE) to eliminate potential effect of systematic bias in the SoilNet data, i.e. we target the matching of the pattern rather than the absolute values.

## 5 Results and discussion

### 5.1 How robust is the standardization of the sensitivity?

One important requirement to identify a uniform $N_0$ was to scale the neutron count rates of the different sensors to a uniform sensitivity level. To that end, a mobile calibrator probe and concomitant measurements with the rover unit were used (see section 4.1.1). However, the comparison between the rover and the stationary CRNS is not entirely straightforward: while the rover is sensitive enough to obtain sufficiently robust statistics in a period of 30 minutes, the same does not apply to the stationary sensors. Hence, the neutron counts from the stationary sensors had to be averaged over a longer period for which we chose a window of 12 hours around the time of sensor collocation. Fig. 3 shows the corresponding results. For each sensor, it shows the standardized count rates versus those of the rover unit during collocation. The bars indicate the standard errors of the mean neutron count rate (within 12 h and 30 min for stationary and rover units, respectively). A line was fitted through the average values, the slope of which represents the sensitivity ratio between rover and the stationary CRNS sensors standardized





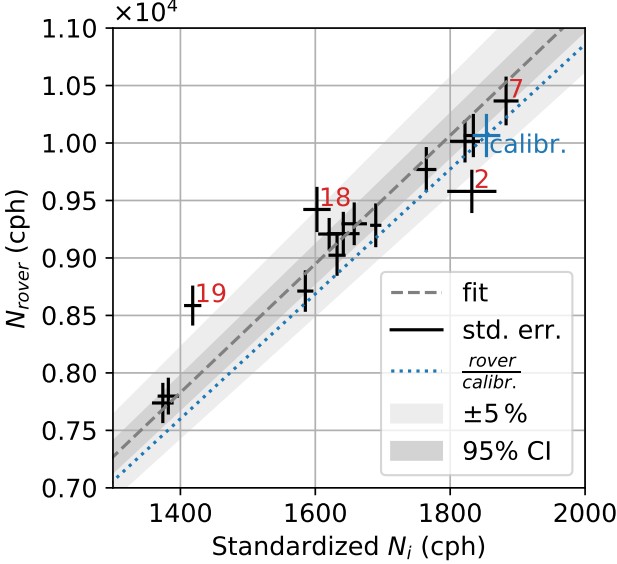

**Figure 3.** Comparison between standardized neutron intensities of the stationary CRNS sensors ($N_i$) and the mobile calibrator (x-axis), and the roving CRNS ($N_{rover}$, y-axis). The solid black lines indicate the standard error of the mean neutron intensities within 30 minutes ($N_{rover}$) and 12 hours ($N_i$, calibrator), respectively; the dashed line represents a linear fit between $N_{rover}$ and $N_i$, the dark shade is the corresponding 95 % confidence interval of that fit; the light shade indicates a deviation of $\pm$ 5 % in $N_{rover}$, the slope of the blue dashed line shows the ratio between $N_{rover}$ and $N_{calibrator}$ when both were collocated; the red numbers highlight the IDs of selected stationary CRNS sensors.

to calibrator sensitivity. Or, in other words, the slope of the fit should correspond to the ratio between the sensitivities of the rover and the calibrator, and amounts to a value of 5.58. The ratio between rover and calibrator count rates, as obtained from the actual collocation of both sensors (shown in blue), amounts to 5.43 - that is about 2.7 % less, which we consider a good agreement. Furthermore, most of the stationary CRNS sensors are very consistent with the rover, particularly given the range

of standard errors. However, three sensors - 2, 18, and 19 - do not exhibit any overlap of the error bars with the 95 % confidence interval of the fitted line which might raise some concern with regard to the integrity of their scaling. Still, a variety of causes could be behind the disagreement. For sensor 19, e.g., the rover had to be placed, due to difficult terrain, at a distance of approx. 5 meters from the sensor which could have a substantial effect given the high sensitivity of the sensors to the near range field. The same applies to sensor 18, which was located close to the rivulet (see Fig. 1) where we can expect very high local soil

moisture gradients. In such a setting, a distance of one meter could already have an effect, and the fact that sensor 19 was located between the rover and the rivulet is consistent with the higher neutron intensity observed by the rover (as it might have been placed on drier terrain). Finally, sensor 2 was affected by a series of erratic count rates just during the time of the roving campaign. The removal of these erratic count rates produced some data gaps, and could have caused some bias. Given these circumstances, it appears that the homogenisation of sensitivity was rather successful. Still, attention should be paid for those

sensors which fall out of line, when we evaluate the results of the $N_0$ calibration.

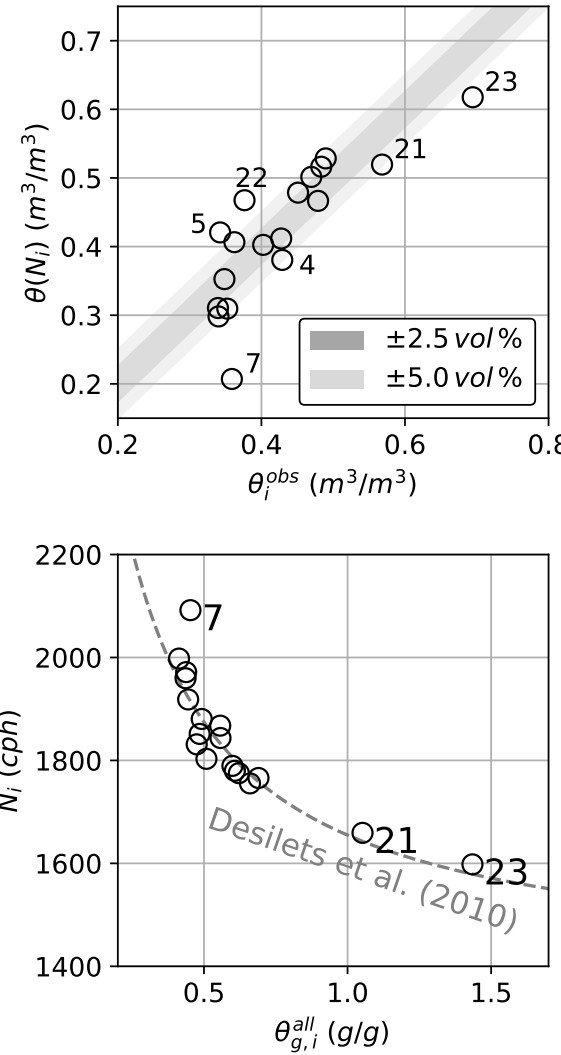

**Figure 4.** Results of the $N_0$ calibration: the upper panel shows the $\theta(N_i)$ as obtained from standardized and corrected neutron count rates $N_i$ versus conventional soil moisture estimates $\theta_i^{obs}$, i.e. footprint-averaged volumetric soil moisture obtained from manual sampling (June 25-26, 2019). The grey shadows indicate a deviation of 2.5 and 5 % in $\theta(N_i)$. The bottom panel provides a different perspective by plotting $N_i$ versus the observed total gravimetric soil water content, including water equivalents from soil organic matter and lattice water; the dashed line indicates the Desilets function, Eq. 1, with a calibrated $N_0$=3723 cph.

## 5.2 Can we find a uniform $N_0$ for the entire study area?

Applying Eq. 1 for converting epithermal neutron count rates to soil moisture requires the parameter $N_0$. It accounts for a large variety of (rather stationary) conditions that affect neutron intensity, mostly sensor sensitivity, hydrogen pools in vegetation,





**Table 2.** Overview of calibration data and results for all involved CRNS sensor. The column headers are as follows. ID: sensor ID, Forest: Percentage of forest area in the footprint; AGB: above-ground biomass (in kg dry matter/m$^2$), weighted average per footprint; N: corrected neutron count rates (in cph), averaged over calibration period; $\rho_b$: soil bulk density (kg/l), weighted average per footprint; $\theta^{obs}$: volumetric soil water content from manual measurements (in m$^3$/m$^3$), weighted average per footprint; $\theta^{eq}$: volumetric soil water equivalent ($\theta^{som}+\theta^{lw}$, in m$^3$/m$^3$), weighted average per footprint; $\theta(N)$: volumetric soil water content as estimated from Eq. 1 (in m$^3$/m$^3$).

| ID | Forest | AGB | N | $\rho_b$ | $\theta^{obs}$ | $\theta^{eq}$ | $\theta(N)$ |
|----|--------|------|------|------|------|------|------|
| 1 | 73 | 18.2 | 1774 | 0.96 | 0.48 | 0.11 | 0.52 |
| 2 | 0 | 0.4 | 1959 | 1.06 | 0.34 | 0.12 | 0.31 |
| 3 | 0 | 0.1 | 1971 | 1.07 | 0.35 | 0.12 | 0.31 |
| 4 | 2 | 0.6 | 1867 | 1.01 | 0.43 | 0.13 | 0.38 |
| 5 | 0 | 0.4 | 1831 | 0.93 | 0.34 | 0.10 | 0.42 |
| 6 | 18 | 4.0 | 1880 | 1.07 | 0.40 | 0.12 | 0.40 |
| 7 | 0 | 0.2 | 2091 | 1.07 | 0.36 | 0.12 | 0.21 |
| 8 | 0 | 0.1 | 1789 | 0.92 | 0.45 | 0.10 | 0.48 |
| 14 | 92 | 20.7 | 1850 | 0.98 | 0.36 | 0.11 | 0.41 |
| 16 | 9 | 2.7 | 1844 | 0.96 | 0.43 | 0.11 | 0.41 |
| 17 | 1 | 0.4 | 1766 | 0.82 | 0.48 | 0.09 | 0.47 |
| 18 | 1 | 0.6 | 1755 | 0.86 | 0.47 | 0.10 | 0.50 |
| 19 | 85 | 13.6 | 1779 | 0.99 | 0.49 | 0.11 | 0.53 |
| 21 | 7 | 2.1 | 1659 | 0.69 | 0.57 | 0.16 | 0.52 |
| 22 | 88 | 21.4 | 1801 | 0.95 | 0.38 | 0.11 | 0.47 |
| 23 | 19 | 4.7 | 1598 | 0.56 | 0.69 | 0.10 | 0.62 |
| 24 | 0 | 0.1 | 1997 | 1.13 | 0.34 | 0.13 | 0.30 |
| 25 | 0 | 0.1 | 1918 | 1.03 | 0.35 | 0.11 | 0.35 |

lattice water, and soil organic carbon. That implies, in turn, that the $N_0$ should be the same for each sensor if we had perfectly
homogenised the neutron intensities of all sensors with regard to these effects (and if, of course, our manual measurements for calibration were perfect, too). Given our corresponding homogenisation efforts (see section 4.1), we hence estimated one single $N_0$ value for all sensors (see section 4.2). Fig. 4 shows the corresponding results of that calibration in which we obtained an $N_0$ of 3723 cph (counts per hour) with a corresponding mean absolute error of 0.047 m$^3$/m$^3$ for volumetric soil moisture. The upper figure panel contrasts volumetric soil moisture estimates from CRNS and manual measurements, while the lower
panel illustrates how well the relation between gravimetric soil water content $\theta_g^{all}$ and neutron intensity $N$ corresponds to Eq. 1, based on the uniform $N_0$ value of 3723 cph. A more detailed overview of the data that is required for the calibration is provided in Tab. 2, which also includes the resulting $\theta(N)$ for each sensor.

Clearly, the agreement is less than perfect. Still, the general pattern suggests that CRNS-based soil moisture estimates obtained from a single $N_0$ can explain a substantial portion of soil moisture variability in the study area ($R^2 = 0.69$), considering





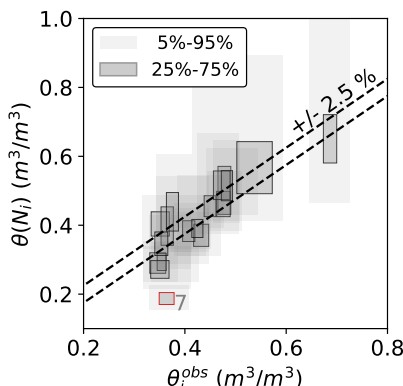 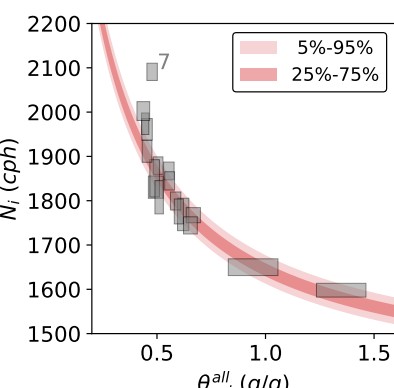 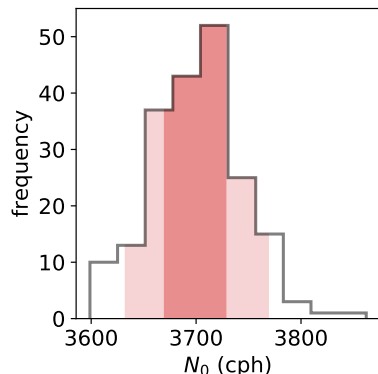

**Figure 5.** Results of the Monte-Carlo-analysis for $N_0$ calibration: the left panel shows $\theta(N_i)$ versus $\theta_i^{obs}$, while the center panel shows corrected neutron intensities $N_i$ over the gravimetric soil water equivalent. The grey shadows and boxes indicate the soil moisture percentiles that result from the 200 realisations of the Monte-Carlo-analysis. The dark grey boxes show the inter-quartile range of soil moisture realisations, and the light grey boxes show the inner 90 % (left panel only). The right panel shows the distribution of all 200 $N_0$ values, highlighting the inter-quartile range in dark and the inner 90 % in light red, respectively. Accordingly, the red shades in the center panel indicate the ranges of the Desilets function when using the inter-quartile range or the inner 90 % of $N_0$ realisations.

475 the large range of soil moisture values from the 0.33 to 0.70 m³/m³. At the same time, some sensors display a remarkable disagreement between $\theta_i^{obs}$ and $\theta(N_i)$, most notably for sensor 7 (-0.15 m³/m³), but also for sensor 22 (+0.09 m³/m³), sensor 5 (+0.08 m³/m³), and sensor 4 (+0.06 m³/m³).

 Before discussing potential reasons for these disagreements, we would like to put the results into perspective, based on the Monte-Carlo-analysis outlined in section 4.3. That analysis aims to get a better idea of the uncertainties involved in the $N_0$

480 calibration (cf. section 4.3). The results of the Monte Carlo analysis are shown in Fig. 5. The figure has the same layout as Fig. 4, but it displays, based on the 200 realisations of the Monte-Carlo-analysis, the uncertainty ranges of $\theta(N_i)$ and $\theta_i^{obs}$ (left panel), $N_i$ and $\theta_{g,i}^{all}$ (center panel), and the resulting distribution of $N_0$ values (right panel).

 As already pointed out, Fig. 5 puts into perspective the disagreements observed in Fig. 4. Or, in other words, Fig. 5 is both encouraging and disappointing.

485 It is rather disappointing to see that the local uncertainty of soil moisture is substantial, as expressed by both the vertical and the horizontal extents of the shades. The vertical shades in the left panel indicate that $\theta(N_i)$ is highly uncertain. There are multiple sources for that uncertainty. Its extent is a location-specific combination of the uncertainties from e.g. the estimation of above-ground forest biomass, the uncertainty of the mean $N_i$ used for the calibration (which was represented by temporal sub-sampling), or the uncertainty of the spatial distribution of bulk density in the sensor footprint. Yet, $\theta_{obs}$, the so-called

490 "ground-truth", bears a considerable degree of uncertainty, too. That is partly caused by the uncertainty of the local soil moisture measurement itself, but more importantly, due to the limited spatial density of the measured profiles. In prospective studies, a systematic sensitivity analysis would be helpful in order to disentangle the different sources of uncertainties.





However, it is encouraging that the estimation of $N_0$ appears quite robust, and that there is no evidence to suggest that $N_0$ is not uniform (except for sensor 7). The 25th and 75th percentiles of $N_0$ amount to 3677 and 3733 cph, respectively, which

corresponds to a range of less than 1.5 % relative to the optimal value of $N_0 = 3723$ cph. The boxes which represent the interquartile ranges of $\theta(N_i)$, $\theta_i^{obs}$, and $N_i$, clearly overlap with the diagonal (Fig. 5, left panel), and mostly with the curve of the Desilets function, respectively (Fig. 5, center panel).

Only for CRNS sensor 7, the box does neither connect to the diagonal nor to the Desilet function. Hence, we cannot assume that the $N_0$ value of 3723 cph is valid for this CRNS location. We currently do not have a satisfactory explanation for that.

Vegetation biomass in the footprint of sensor 7 is comparably small and its uncertainty cannot, by any means, explain the level of disagreement. The same applies to the uncertainty of water equivalents from soil organic matter and lattice water. We also assume the standardization of sensitivity for sensor 7 to be successful, based on Fig. 3 in which the neutron intensity as observed by sensor 7 corresponds very well to the rover measurement. Then again, the reference soil moisture $\theta_7^{obs}$, might not be sufficiently representative for the entire footprint of sensor 7 due to a lack of manual measurements, especially in the

western direction. That lack was mainly due to access restrictions (see Fig. 1) and technical issues which led to the loss of manual FDR profiles at five locations near sensor 7. While manual measurements are scarce around some other CRNS sensors, too, the constellation around sensor 7 might be specifically unfortunate. That is because the soil in that area generally appears to become drier towards the west, while no manual measurements were available in this direction, resulting in an omission of this potentially important soil moisture gradient. A preliminary analysis of the roving data from June 2019 had also indicated

relatively dry conditions along the road in the south-west of sensor 7 (see Fig. 10 in Fersch et al., 2020a). However, CRNS-based soil moisture estimates along roads are also prone to underestimation due to the so-called road-effect (Schrön et al., 2018). In summary, it would be reasonable to suspect that $\theta_7^{obs}$ is lower than the actual average soil moisture in that area – maybe even up to the observed discrepancy of 0.15 m$^3$/m$^3$. Unfortunately, though, we do not have hard evidence to support this hypothesis. In fact, it would be at least surprising if location 7 would turn out to be so much drier than the other (rather

dry) locations north and south of it. On the basis of this discussion, we decided to exclude sensor 7 from further analysis in the present study.

### 5.3 Spatial and temporal patterns in CRNS-based soil moisture

The next step of our analysis is to use the observed neutron intensities, together with the uniform $N_0$, in order to retrieve time series of volumetric soil moisture for the remaining 17 locations and over the entire study period. For that purpose, we first

compute a 12-hours moving average of the corrected neutron intensities which are then converted to volumetric soil moisture using Eq. 1. Fig. 6 shows the corresponding results, together with the meteorological forcing in terms of precipitation and potential evapotranspiration.

Overall, we see very strong temporal dynamics of wetting, drying, and re-wetting that correspond well with the cumulative difference between precipitation and reference evapotranspiration (as indicated by the red line in the top panel of Fig. 6).

Torrential rains of about 150 mm from May 20-22 marked the start of the campaign, with another 50 mm of rain following until May 29. During that period, many locations exhibited an apparent volumetric soil moisture higher than soil porosity (as

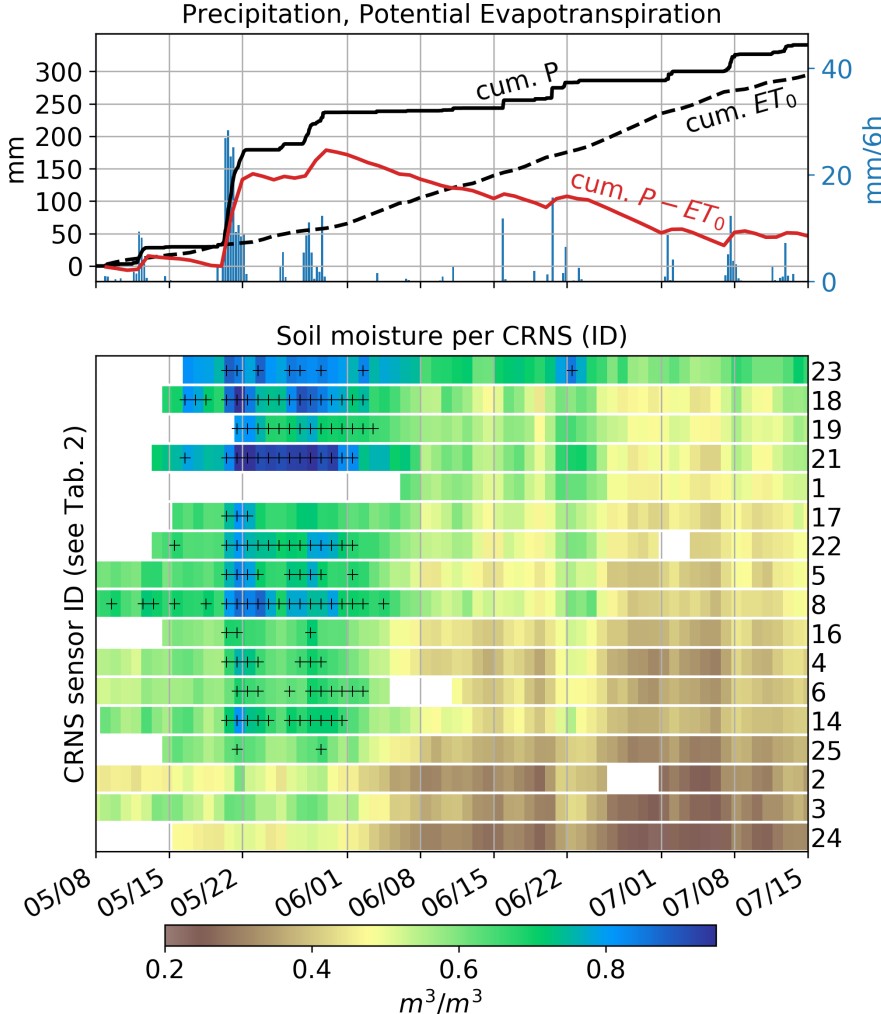

**Figure 6.** Time-series of precipitation, reference evapotranspiration, and estimated soil water content. Upper panel: cumulative precipitation P (mm) and precipitation depth (mm) per 6 hour intervals (blue bars), reference evapotranspiration $ET_0$ (based on FAO, 1998, in mm), and the difference $P - ET_0$ (in mm). Bottom panel: soil moisture $\theta(N_i)$ as estimated from corrected and standardized neutron intensities $N_i$ of each CRNS sensor $i$; rows are sorted in descending order, from top to bottom, based on the average soil water content during the last week of the campaign (July 8-15). A white space indicates a period of missing data, the black "+" signs indicate periods in which the apparent volumetric soil moisture exceeds soil porosity, probably due to ponding.

marked by the black + signs in Fig. 6), i.e. higher than full saturation. That is in line with the observation that large parts of the study area were affected by inundation and ponding in the last week of May 2019. The following drying period from end of May until the end of the campaign on July 15 was repeatedly interrupted by substantial rainfall events of more than 10 mm (e.g. on June 15 and 20, July 1, 7 and 12). In total, June marks the transition from extremely wet to much drier conditions: from






July 1 to 30, soil moisture dropped by 0.2 - 0.3 $m^3/m^3$ for each CRNS location (which corresponds to a decrease between 25 to 50 percent relative to the soil moisture on June 1).

The rows in the bottom panel of Fig. 6 have been sorted in descending order, from top to bottom, based on the average soil water content during the last week of the campaign (July 8-15). From that arrangement, some patterns in space and time
emerge. Location 23, located in a large patch of peat soil, stands out with very high soil moisture values that are permanently above a value of 0.5 $m^3/m^3$ (with an average of 0.71 $m^3/m^3$). At the other end, there is a group of relatively dry locations in the western parts of the study area (locations 2, 3, 4, 24, and 25), and, less pronounced, but still rather dry, along the eastern slopes towards the water divide of the catchment (locations 6, 14, and 16). In-between, we find a series of locations with intermediate soil moisture levels and pronounced wetting and drying dynamics over the study period, most of them strung
along the central valley bottom of the Rottgraben (locations 1, 5, 17, 18, and 21) as well as a drainage line from the eastern slopes (location 19). Location 8 is more peculiar: it is very close to the driest locations, starts off very wet, but also exhibits pronounced drying dynamics.

We also have to keep in mind the results from section 5.2, which indicated the large local uncertainty of $\theta(N)$. Hence, any ranking that is based on average soil moisture values should be interpreted with care. Take location 22, for instance: Fig. 4
suggests that $\theta(N)$ is probably overestimating soil moisture in this location which is supported by the uncertainty range shown in Fig. 5. The dominant source of uncertainty at location 22 is most likely the above-ground forest biomass in the near range of sensor 22. Hence, location 22 could well be drier than shown in Fig. 6 by around 0.1 $m^3/m^3$. Examining, however, the way a CRNS location changes rank over time - relative to other locations - should be less affected by such local (but rather static) uncertainties. From a hydrological perspective, it might be informative to examine such changes in rank as they may indicate
changes in governing hydrological processes. Just as an example, location 21 was by far the wettest location in the last week of May, but only the fourth wettest location in the last week of the campaign. Same as location 23 (the wettest location on average), location 21 is characterised by peaty soils with low bulk density and high soil moisture, but it falls dry much faster. We hypothesize that during the heavy rainfall in May, location 21 collected near-surface inter-flow from the nearby eastern hillslopes which, in combination with a local impermeable layer below the peat soils, led to a local accumulation of water.
However, location 21 might not be influenced as much by the shallow aquifer as location 23, so it dried faster. At the moment, such hypotheses remain untested. Yet, they illustrate how such a dense network of CRNS sensors in a heterogeneous landscape could help us to formulate and test hypotheses on governing hydrological processes.

## 5.4 Spatial interpolation of soil moisture

This is the first study that aims at using CRNS data to coherently represent the spatial heterogeneity and temporal dynamics of
soil water content at the catchment scale. With the term "catchment scale", we refer, in the present study, to the variation of soil water content at a scale of tens to hundreds of meters, i.e. the variation *within* and *between* CRNS footprints. In section 4.4, we outlined two interpolation models which are both based on Ordinary Kriging: an "unconstrained" and a constrained" model, which can be applied to any given point in time within the observation period.

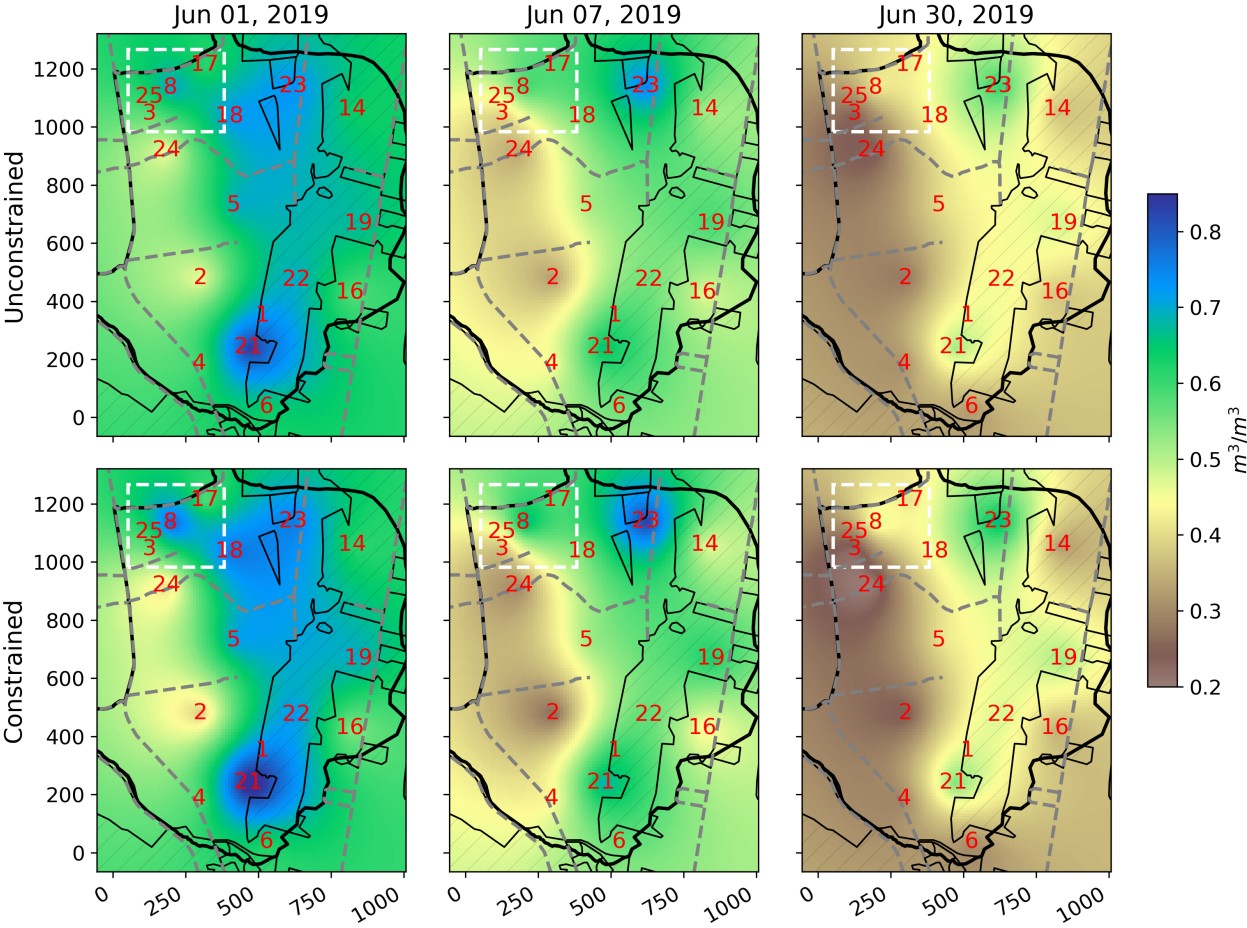

**Figure 7.** Daily average soil moisture for three different dates (left: June 1, center: June 7, right: June 30, 2019) and for the unconstrained and the constrained interpolation model (top and bottom row, respectively). The red numbers mark the locations and IDs of the CRNS sensors used for the interpolation; the wide solid black line marks the catchment boundary, the black hatches show forested areas, the grey dashes show roads, and the white dashed rectangle in the north-west indicates the extent of the SoilNet. See section 3.7 on the usage of OSM basemap data (© OpenStreetMap contributors, 2020, distributed under ODbL license).

Fig. 7 shows the resulting soil moisture maps for three different dates which differ with regard to the average soil moisture:

very wet conditions on June 1 (two days after the continued rainfall in May 2019 had ended); intermediate conditions on June 7 (after a week of drainage and evapotranspiration); and relatively dry conditions on June 30 (after another three weeks of drying).

For both interpolation models, the maps illustrate more intuitively what we have already learned in the previous section (section 5.3: soils are becoming considerably drier from the valley bottom of the Rottgraben towards the western parts, and,

less pronounced, but still obvious, towards the eastern hillslopes. Furthermore, the locations of sensor 19 (at a drainage line)



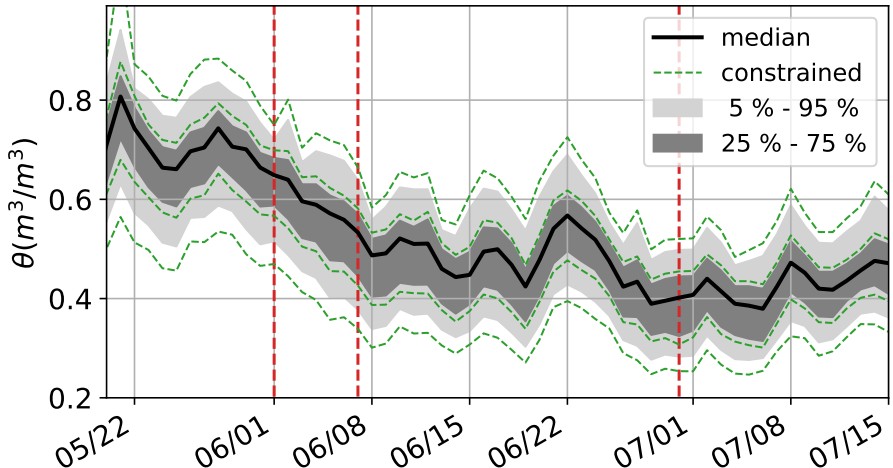

**Figure 8.** Spatio-temporal distribution of soil moisture in the catchment as a result of interpolation: the solid black line shows the catchment median of soil moisture as computed from the unconstrained interpolation model, the dark grey shadow shows the inter-quartile range, and the light shadow the inner 90 %. The dashed green lines indicate the corresponding percentiles (5th, 25th, 75th, and 95th) as computed from the constrained interpolation model (the median is not shown in addition as it is almost identical). The vertical red dashed lines indicate the dates used in Fig. 7.

and 8 (in the SoilNet area) are rather wet. The maps also illustrate well how the drying of soils progresses in space in the course of June.

At the same time, Fig. 7 also demonstrates the differences between the two interpolation models: as expected, the constrained model enhances horizontal soil moisture gradients, as visible from the increased contrasts in the bottom panel maps.

At this point, we should reiterate that the interpolation is a necessary step to infer the average soil moisture for the entire catchment: a simple averaging of soil moisture, as obtained from neutron intensities, would not account for the uneven spatial distribution of sensors. The interpolation of soil moisture, in turn, not only allows us to compute an areal average, but to capture the actual frequency distribution of soil moisture values in the catchment for any point in time. Such an attempt is illustrated by Fig. 8. The shaded areas (dark for the inner 50 % and light for the inner 90 %) are based on the unconstrained

interpolation. The dashed lines indicate the change of the corresponding percentiles (5th, 25th, 75th, and 95th) when using the results of the constrained interpolation. The catchment median for the constrained interpolation is almost identical for the constrained interpolation and hence not additionally shown. Obviously, the difference in soil moisture between the interpolation methods particularly differs for extreme percentiles. That is consistent with our expectation: in the constrained interpolation, dry locations tend to be drier, and wet locations wetter as compared to the unconstrained model.

What remains, however, is the question about the validity of the interpolation models. There is no independent catchment-wide soil moisture distribution that could serve as a reference for a validation. But we do have the SoilNet observations in the





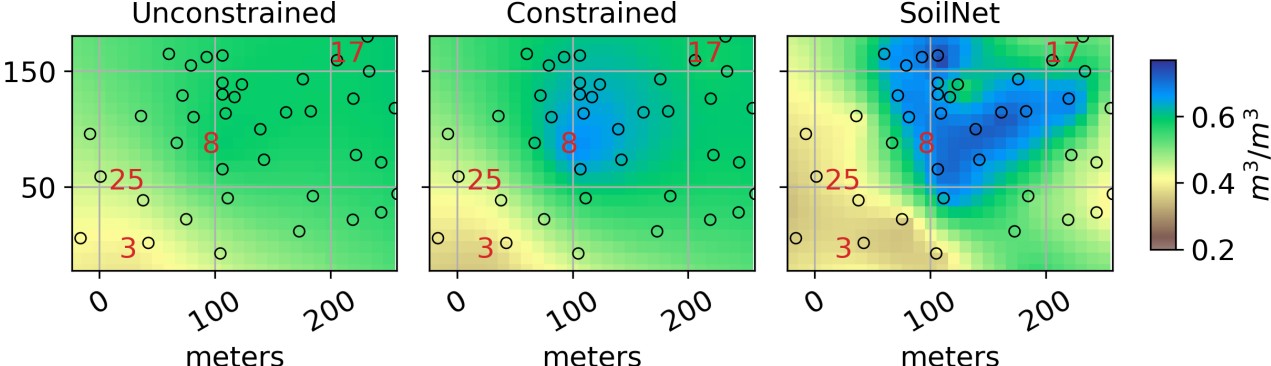

**Figure 9.** Daily average of root-zone soil moisture in the SoilNet area on June 7, 2019, as obtained from the unconstrained interpolation (left panel), the constrained interpolation (center panel), and the interpolated SoilNet measurements (right panel). The black circles indicate the locations of the SoilNet nodes, the red numbers indicate the locations and IDs of the CRNS sensors within the SoilNet bounding box.

area that is marked by the white dashed rectangle in Fig. 7. In the following section, we will hence compare the soil moisture patterns as obtained from both interpolation models to the soil moisture patterns inferred from the SoilNet.

### 5.5 Comparison of spatial patterns against SoilNet observations

The area of the SoilNet is particularly suited to examine whether the constrained model is in fact superior to the unconstrained model in representing soil moisture gradients at the catchment scale: first, of course, the SoilNet itself allows, based on its high sampling density, to capture soil moisture variability in space. Second, the density of CRNS sensors is also high in that area: the 150 m radius of six sensors (3, 8, 17, 18, 24, 25) substantially overlaps with the SoilNet area. Third, according to the data of the FDR-sensors, the spatial heterogeneity of soil moisture in the SoilNet area is quite pronounced, with generally

drier soils towards the West, together with a wet anomaly at the center of the SoilNet. The second and the third condition are both required for the constrained model to actually make a difference: a sufficient overlap of footprints together with distinct horizontal soil moisture gradients.

In section 4.5, we have outlined how the SoilNet observations can actually be compared to the interpolated $\theta(N_i)$: we vertically average the FDR-measurements, horizontally interpolate these vertical averages to the same grid as used before, and

then compare those parts of the grid that fall inside the spatial bounding box of the SoilNet. Fig. 9 shows an example for June 7, 2019, i.e. for intermediate wetness conditions. The spatial window corresponds to the dashed white box in Fig. 7. The left panel shows the soil moisture pattern according to the unconstrained model, the center shows the constrained model, and the right panel the pattern according to the SoilNet. The SoilNet shows a wet area in its central part that tends to extend to the north and the northeast. It features a pronounced progression towards drier conditions in the west, southwest and southeast. It

appears that this soil moisture gradient from the center to the west and south is well captured by the constrained interpolation



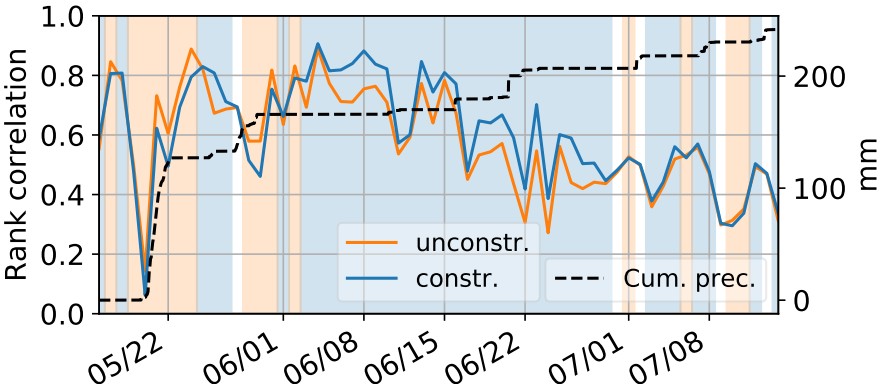

**Figure 10.** Correlation of interpolation models with the SoilNet observations. The orange (blue) line shows the rank correlation coefficient between the daily average soil moisture pattern of the SoilNet and the one of the unconstrained (constrained) interpolation of CRNS-based measurements. Blue and orange backgrounds highlight days in which one correlation model outperforms the other. The dashed black line shows the cumulative precipitation over the study period.

model, particularly in comparison to the unconstrained model. Both models fail to reproduce the dry region in the eastern part of the area.

In order to formalize the comparison, Fig. 10 shows Spearman's rank correlation between the spatial soil moisture patterns (daily averages) obtained from the interpolation models (constrained and unconstrained) and the SoilNet. Apparently, the

constrained model is superior, i.e. more similar to the SoilNet, for the majority of the study period, namely all June. Only for the very wet beginning of the campaign (until early June), the unconstrained model seems to largely outperform the constrained one. That period, however, has to be interpreted with care since it was governed by substantial ponding, which obviously cannot be captured by the SoilNet sensors, while being detected by the CRNS network. Hence, that period might in fact be unsuited to evaluate the superiority of any of the interpolation models. We can also observe a general decline of correlation between the

CRNS-based interpolation products and the SoilNet towards the (drier) end of the campaign: in July, the differences in rank correlation between both interpolation models are rather negligible. Furthermore, rainfall events tend to reduce the correlation which is a well-known issue when using subsurface sensors as a reference: until the infiltration of a rainfall advances to the upper SoilNet sensor (in 5 cm depth), the SoilNet does not register the event while the CRNS sensors immediately react to the additional water at the soil surface, even more while it is close to the surface (Schrön et al., 2017; Baroni et al., 2018; Scheiffele

et al., 2020).

In summary, it appears that the constrained interpolation model is more successful in capturing horizontal soil moisture gradients, as represented by the SoilNet, than the unconstrained model. Yet, we should be aware that the reliably of the SoilNet itself as well as its use as a reference observation certainly has its limitations. In future studies, the usefulness of the available roving data, as published by Fersch et al. (2020a), should be examined in the context of such benchmarking experiments.





## 6  Summary and conclusions

This study is the first attempt to analyse a comprehensive observational campaign that took place the summer of 2019, in which a pre-alpine catchment with an area of $1\,km^2$ was covered, for two months, by a dense network of 24 stationary CRNS sensors. Based on the recently published data set (Fersch et al., 2020a), we investigated the potential to homogenise observations from a heterogeneous sensor network in a heterogeneous landscape under heterogeneous hydro-meteorological conditions. That homogenisation is a necessary step towards a uniform retrieval of soil moisture, and hence towards characterizing the spatio-temporal heterogeneity of the study area with regard to soil moisture. In order to coherently and comprehensively represent that heterogeneity across the catchment, we interpolated the CRNS-based soil moisture estimates in space. For that purpose, we suggested a new technique that combines the concept of spatial interpolation with the idea of a geophysical inversion – with the aim to maximise the consistency of spatial soil moisture patterns with the observed neutron count rates. In the following, we will highlight the main lessons learned along these steps.

### 6.1  Homogenisation and $N_0$ estimation

The homogenisation aims at making observed neutron count rates comparable in space and time, or, in other words, to quantify and eliminate all effects that are not caused by soil moisture, so that the remaining signal informs us about the variability of soil moisture in space and time. The first step was to standardize the neutron intensities as observed by the different sensors with different sensitivities. That was achieved by collocating most of the CRNS sensors with a so-called calibrator probe. Based on the ratio of neutron counts of the collocated sensors during an integration interval of at least one day, we inferred sensitivity factors. Between the various sensor types, these factors varied over an order of magnitude - as the first generation of CRNS sensors (CRS 1000) was operated abreast with the latest high-sensitivity sensor types (NeuSens Dual). We verified the consistency of that standardization by using an independent mobile sensor from a CRNS rover. That sensor had been collocated with 16 stationary CRNS sensors for at least 30 minutes in the period from June 25 to 26, 2019, and we found the agreement between the standardized neutron intensities and the rover to be high. Only three sensors (2, 18, 19) exhibited a noticeable disagreement which, however, could also be explained by strong local soil moisture gradients in the near field of the sensors. Altogether, the calibrator-based standardisation and its rover-based verification turned out to be an important basis for this study. In the future, it would be very helpful to systematically collect and publish sensitivity measurements and sensor inter-comparisons which highlight the variability of sensitivity between sensor types, but also within the same sensor type. Ideally, such studies should refer to a common standard. That way, future inter-comparisons of CRNS records from heterogeneous networks could be facilitated without the additional (and substantial) effort of collocating a calibrator probe.

The next step in homogenisation involved routine procedures to account for the atmospheric effects which were assumed to be dynamic in time, but constant in space: barometric pressure, air humidity, and the incoming neutron flux. In turn, the effects of other hydrogen pools in the soil and at the surface were assumed to be variable in space, but constant over time, namely vegetation biomass, soil organic carbon, and lattice water. While the effect of lattice water was rather negligible, soil organic carbon showed a strong variation between the organic soil around sensors 21 and 23, and the other sensors that were mainly





surrounded by mineral soils. Vegetation biomass differed by about two orders of magnitude between grassland and forest. Yet, much more important with regard to the uncertainty of the CRNS-based soil moisture estimation was the small scale variability

of biomass within the forest due to highly heterogeneous species and age structures.

After accounting for these various effects, we estimated a uniform $N_0$ value of 3723 cph for a set of 18 CRNS locations, at a mean absolute error of 0.047 m³/m³ with regard to volumetric soil moisture during the calibration phase. The results suggest that CRNS-based soil moisture estimates, as obtained from a single $N_0$, can explain a substantial portion of soil moisture variability in the study area ($R^2 = 0.69$). In a Monte-Carlo-analysis, we repeated the $N_0$ estimation 200 times with assumptions

on various sources of uncertainty. It turned out that the local uncertainty of $\theta(N_i)$ can be high, for example in locations that are affected by forest biomass and high local gradients of soil organic carbon. The uncertainty of $\theta_i^{\mathrm{obs}}$, as obtained from the ground truthing campaign, can be high, too, as a result of the limited sampling density for soil moisture and bulk density. Still, the estimation of a single $N_0$ appeared to be robust, i.e. the variation of $N_0$ over all realisations of the Monte-Carlo-analysis was small. Given that the potentially large local uncertainty of both $\theta(N)$ and $\theta_{obs}$ is something we need to accept, at least in

the present study context, the estimation of one single $N_0$ at least keeps us from overemphasizing specific features in the data of individual sensors (in case we would calibrate each sensor individually). We hence consider the estimation of a joint $N_0$ as an important step towards a methodological generalisation – a generalisation that is required if we aim at upscaling the CRNS method from individual footprints to larger-scale networks or clusters. For that purpose, we not only require transferable and scalable methods to reliably quantify the variables of interest – vegetation biomass, bulk density, soil organic matter content

and, of course, soil moisture –, but we also have to better understand which factors govern the remaining uncertainties.

### 6.2 Spatio-temporal patterns of soil moisture

Using our $N_0$ estimate to retrieve time series of volumetric soil moisture at each CRNS location, we were able to observe pronounced dynamics of wetting, drying, and re-wetting for all locations. Torrential rains marked the start of the campaign. In the last week of May, $\theta(N)$ even exceeded soil porosity at many locations. That indication of inundation and ponding is well in

line with visual observations of the study area during that time. The following drying period in June marks the transition from extremely wet to much drier conditions (soil moisture dropped by values between 0.2 and 0.3 m³/m³), repeatedly interrupted by substantial rain events. Plausible spatial patterns emerged, patterns that only became apparent by the dense CRNS coverage of a heterogeneous study area: regarding wet conditions, location 23 stood out with very high soil moisture values. Relatively dry locations were found in the western and eastern parts of the study area. Locations with intermediate soil moisture levels

and pronounced wetting and drying dynamics were mostly strung along the central valley bottom of the Rottgraben. We also found that – if we ranked the locations according to their average soil moisture – the ranking of some stations remained rather stable, while others showed pronounced changes. We suggest that such changes might be informative with regard to changes in governing hydrological processes.

In summary, $\theta(N)$ at the footprint scale can already convey fundamental insights into the variation of soil moisture in a

landscape, if multiple sensors are operated in a dense network. In order to formalize and extend that view, we interpolated $\theta(N)$ in space, aiming at a representation of soil moisture variability at a scale of tens to hundreds of meters (within and



between CRNS footprints). As a result, we can compute, at a given day, the average soil moisture of the $1 \, \mathrm{km}^2$ catchment, or the soil moisture frequency distribution inside the catchment, both of which could be useful to the validation of hydrological models or remote sensing products at various scales.

In this study, we specifically compared two interpolation models which are based on Ordinary Kriging: the "unconstrained" model provides a straightforward interpolation of the $\theta(N)$; the "constrained" model also interpolates, but adjusts the values of $\theta(N)$ at the CRNS locations (interpolation nodes) subject to minimising the disagreement between the observed neutron count rates and the interpolated soil moisture field. In essence, the constrained model is similar to what is generally referred to as a "geophysical inversion", i.e. the identification of parameters $\mathbf{p}$ in a model $m$ by means of inverse simulation: typically,

the observed variable is obtained from the modelled variable by means of a physically-based forward operator, and $\mathbf{p}$ is optimised by minimising the disagreement between the simulation of the observed variable and the observation itself. That way, further spatial information can be obtained from volume-integrated observations by using our physical understanding of both the observed system and the observation technique itself. Admittedly, our "constrained" approach does not entirely qualify as a "geophysical inversion": first, we used a geostatistical instead of a physically-based model to describe our notion of a

spatially continuous soil moisture variation at a specific scale. Second, we used a rather heuristic implementation of a forward operator $\mathscr{H}$ instead of a neutron transport model. Despite these limitations, we presented an entirely new approach to use the horizontal sensitivity pattern of the CRNS sensor in order to constrain the spatial pattern that results from an interpolation. We demonstrated that the constrained interpolation model is able to emphasize horizontal soil moisture gradients at the scale of decameters to hectometers, and hence leads to drier and wetter conditions at the tails of the soil moisture distribution in the

catchment, while preserving its median values. These effects are certainly relevant in the context of hydrological modelling as runoff generation is typically assumed to take place in the wettest parts of a catchment. Using the SoilNet observations in the north-west of the catchment, we verified that the constrained interpolation model is in fact superior in representing the spatial heterogeneity of soil moisture as observed by the SoilNet, although that superiority is not entirely unanimous, and the representativeness of the SoilNet measurements is also questionable to some extent. Moreover, our verification did not extend

beyond the SoilNet coverage. As a proof of concept, though, the constrained model could lead the way towards a general framework to regionalise measurements obtained from dense CRNS networks. Using Ordinary Kriging as a vehicle for the interpolation part was an admittedly arbitrary decision. However, the concept is open for any type of parametric regionalisation or interpolation technique. It would also be possible to establish statistical relationships between landscape attributes and soil moisture, or to use a distributed hydrological model. Any relevant parameters of such relationships or models could then be

inversely adjusted in order to maximise the agreement with observed neutron intensities. This generic property makes the proposed approach particularly attractive, as it can be adjusted to specific situations in terms of available data and methods.

Benchmarking the performance of such alternative model formulations within the "constrained interpolation" framework will be an important activity for prospective research. The concept will be tested with new data from additional campaigns with very dense CRNS networks: from September to November 2020 in the TERENO site Wüstebach (a small headwater

catchment in the Eifel mountains, western Germany), and from August 2910 to August 2020 in a heterogeneously managed agricultural test site near Potsdam (eastern Germany). Such benchmarking activities should also include additional reference

data in order to evaluate the model performance, namely CRNS roving transects. Furthermore, we suggest to examine the validity of the "heuristic" forward operator in experiments with synthetic 3-D soil moisture data, in which the simulations of a physically-based neutron simulation models such as URANOS (Köhli et al., 2015) are compared against the results obtained
by the forward operator that was suggested in this study.

To conclude, this study is admittedly rich in assumptions and fairly arbitrary decisions; that involves the choice of interpolation techniques and their parameters, the definition of a forward operator, assumptions on the spatial representativeness of measurements with regard to various variables – e.g., bulk density, soil moisture, vegetation biomass, soil organic carbon –, the sources and ranges of uncertainty, and many more. Certainly, this arbitrariness needs to be brought down, step by step, in
future research efforts. But despite these degrees of freedom, we have demonstrated how a comprehensive analysis of a such dense CRNS network can be robust and informative with regard to the properties of our data, and the properties of our study area.

*Code and data availability.*

The data used in this study are accessible at EUDAT (https://doi.org/10.23728/b2share.282675586fb94f44ab2fd09da0856883,
Fersch et al., 2020b). An extensive description is provided by Fersch et al. (2020a). The data analysis presented in this study is documented in a Jupyter notebook that is available at GitHub (https://github.com/cosmic-sense/jfc1-analysis-hess, Heistermann, 2020).

*Author contributions.*

MH and TF designed the study, MH wrote the manuscript, TF, MS and SO co-designed the study and co-wrote the
manuscript. SO had proposed the concept of a dense CRNS network.

*Competing interests.* The authors declare that they have no competing interests.

*Acknowledgements.* This research was funded by the Deutsche Forschungsgemeinschaft (DFG, German Research Foundation) – project 357874777 of the research unit FOR 2694 "Cosmic Sense". The TERrestrial Environmental Observatory (TERENO) Pre-Alpine infrastructure is funded by the Helmholtz Association and the Federal Ministry of Education and Research. Base map data copyrighted OpenStreetMap
contributors and available from https://www.openstreetmap.org.



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
