# Peer review of "Spatio-temporal soil moisture retrieval at the catchment-scale using a dense network of cosmic-ray neutron sensors"

_Hydrology and Earth System Sciences, 2021_

## Author Comment (AC1)

**Interactive Discussion: Author Response to Referee #1**

**Spatio-temporal soil moisture retrieval at the catchment-scale using a dense network of cosmic-ray neutron sensors**

Maik Heistermann et al.

*Hydrol. Earth Syst. Sc. Discuss.,* `doi:10.5194/hess-2021-25`
* * *
**RC:** *Reviewer Comment*,     AR: *Author Response*,     ☐ Manuscript text

Dear Referee,

we would like to thank you very much for your positive comments and constructive suggestions to our manuscript. We very much appreciate the time and effort that you have invested in your report.

Based on your report, we have already started to revise our manuscript. Please find below our detailed responses to all the points you have raised in your report. We will continue to address these points, and are confident that the manuscript will substantially improve as a consequence. Yet, the final implementation of changes will also depend on another referee report that is still to be submitted in the interactive discussion.

Kind regards,
Maik Heistermann
(on behalf of the author team)

**1.1. General comments**

**RC:** *[...] my main concerns are with the presentation quality of the paper: the length and structure of the manuscript could be substantially improved (see specific comments). Therefore, I would like to suggest major revisions.*

AR: We will restructure and shorten the manuscript according to the following suggestions of the referee (see our responses below).

**1.2. Structure**

**RC:** *While this is a discussion paper, I would encourage the authors to be more concise and structured when presenting their findings (examples are given below). There are sections which seem to be excessively long. That would hopefully help address another issue, which is the often "downplaying" of the study findings (examples also given).*

AR: We will shorten the manuscript were possible. We also appreciate the referee's concern on "downplaying" the results. We will certainly aim to remove statements that appear unnecessarily critical towards the outcomes of our own study. Yet, we consider it important to clearly and unambiguously emphasize any limitations of our

results, especially as this is the first attempt of inferring spatio-temporal representations of soil moisture from a dense CRNS network, and as we had to make a number of assumptions. We hope that the referee will agree.

**RC:** *Moreover, during the discussion the implications of the findings and potential benefit of the application of these novel analyses are shown/discussed but in a scattered manner throughout "Results and discussion" section. I would recommend condensing them in a subsection within "Summary and conclusions". Below I outline specific examples:*

AR: We suggest to split the former section 6 (Summary and conclusions) to two sections: the new section 6 would be "Lessons learned", in which we highlight the main lessons with regard to homogenisation/$N_0$-estimation and the retrieval of spatial patterns from the dense network; the new section 7 would be a very brief "Conclusions" section. We think that this is also in line with what you suggested in your comment 1.7.

**1.3. Presentation of aims and objectives**

**RC:** *In Section 1.2. Aims and objectives: research questions and specific objectives of the study are presented between lines 65-95 and surely could be condensed. If you would wish to keep all the text, separate clearly in "aims and objectives" and "justification". That would help structuring further the manuscript.*

AR: We agree that it would improve the readability if we moved the specific objectives of the study to a dedicated subsection (1.3: Specific objectives). We will also revise the specific objectives based on our response to comment 1.6. Finally, we will move the overview of the manuscript structure (ll. 96-102 of the original manuscript) to a separate subsection (1.4: Manuscript structure).

**1.4. Section 4.4**

**RC:** *Section 4.4. has effectively three subsections "Unconstrained model", "Constrained model" and "Forward operator". Please give these index numbers 4.4.1., 4.4.2 and 4.4.3. (similar to what you have done in section 4.1.). Also, lines 319-340 from subsection Unconstrained model: please try to shorten and restructure. Only in line 327 you say what you have actually done in the study and before that you give several examples (319-327). I would encourage you to first say what you did and then give the examples.*

AR: As suggested by the referee, we will number the subsections "Unconstrained model", "Constrained model" and "Forward operator". We will also shorten the presentation of the unconstrained approach. However, we would like to keep the order in which it is presented. That is because the terminology and notation with regard to $m$, $\mathbf{p}$, and $m(\mathbf{p})$ is, in our opinion, best introduced by example.

**1.5. Section 5.2.**

**RC:** *Section 5.2. between lines 473-484. – text can be shortened, and arguments presented more concisely. For example, line 472 "Clearly, the agreement is less than perfect. Still, the general pattern suggests", can be rephrased to "While the agreement is not perfect, the general pattern suggests...".*

AR: We will adopt the referee's suggestion for the first sentence of that paragraph, and will further shorten the text that originally extended from ll. 473-484.

**1.6. Align section 5 with specific objectives**

**RC:** *In general, in Section 5: try to restructure main discussion points (5.1. to 5.5.) in a way that it follows the flow of your aims and objectives.*

**AR:** We agree that it would be helpful to better align the structure of the results section with the study objectives/research questions. In fact, the structure of section 5 already aligns quite well with the aims outlined in the introduction. Yet, the correspondence between the individual sub-sections of section 5 to the aims should be more explicit. To that end, we will adjust both the aims *and* the structure of section 5: the first research question in the new section 1.3 (see comment 1.3) will be split in two; the former sections 5.4 and 5.5 will be put in one section with two sub-sections. The result could look like this:

    RQ1 *How robust is the standardization of the sensitivity?* - addressed in section 5.1

    RQ2 *Can we find a uniform $N_0$ for the entire study area?* - to be addressed in section 5.2

    RQ3 *What do the differences between the soil moisture estimates at the CRNS footprint scale tell us about soil moisture patterns at the catchment scale?* - to be addressed in section 5.3

    RQ4 *How does soil moisture vary within and between the sensor footprints?* - to be addressed in section 5.4, with subsection 5.4.1 being "Spatial interpolation of soil moisture", and subsection 5.4.2 being "Comparison of interpolation results against SoilNet observations"

**1.7. Summary and conclusions**

**RC:** *Section 6 Summary and conclusions (which are then called "main lessons learned along these steps") needs to be more concise. Shortly introduce the "main lessons" and then expand on them in 6.1. and 6.2.*

**AR:** Please refer to our response to comment 1.7.

**1.8. Practical guidance for future dense CRNS networks**

**RC:** *Section 4.3. is particularly good, as uncertainties have been very well acknowledged and explained. Moreover, the limitations of the study are thoroughly and openly discussed. However, I encourage the authors to include a short comment in discussion on the applicability and reproducibility of the set-up (of dense CRNS networks) and perhaps provide examples of what would be the minimum number of CRNS probes to be included in a similar set-up in order to assess CRNS SWC spatiotemporal variability in a reliable way at the catchment (in this case 1 km2) scale.*

**AR:** The referee suggests to discuss requirements to the design of dense CRNS set-ups in future experiments. While this is an important comment, it becomes apparent that the requirements strongly depend on the specific aims of such experiments. Specifically, we will always face the trade-off between coverage (seek for adjacent footprints) and resolution (seek for strong overlap). Ideally combining the design of such a network with a-priori knowledge on the spatial variability of soil moisture in the target area is, in our opinion, the key to better resolve that trade-off. That is certainly a focus of our future research efforts, and we will attempt to briefly address this issue in the conclusions section.

**1.9. Figure 1**

**RC:** *Figure 1: In the legend you refer to "Climate gauge" and in caption to "Climate station" and in the text (line 169) you use the term meteorological station. Please choose one term only. Besides, from the figure it does not become apparent where are the peaty soils within this catchment are located. From the text we understand that location 23 is on peaty soils, but what is the extend of these? Would it be possible to include the soils on this map too? May be in a separate panel? Finally, the manual samples locations: they seem to be very close to the CRNS sensors, which is ok, but visually it is impossible to distinguish*

*how many you had around each CRNS probe. Could you include information on their number (e.g. in Table 1)?*

AR: We will use the term climate gauge both in the figure, the caption, and the main text. With regard to highlighting the areas of predominantly organic soils, we still have to find an optimal solution: we will either include it in Figure 1, or, as suggested by the referee, in Table 1 (see comment 1.13).

With regard to the manual sampling locations close to the CRNS sensors, these were always five. Details were explained in Fersch et al. (2020). We will add a sentence to the caption of Fig. 1 to emphasize that fact again.

**1.10. Figure 3**

RC: *Figure 3: Is it correct that the rover can sense neutron intensity between 7000 and 11000 cph? (this is just a question, rather than an improvement suggestion)*

AR: Yes, that was the observational range in the present study context, using the UFZ roving CRNS.

**1.11. Figure 5**

RC: *Figure 5: great figure. Could you include the legend also on the right panel for consistency? Also, optionally name the panels as a),b) and c), this will help you shorten the text related to that figure.*

AR: As suggested, we will also add a legend to the right panel of this figure. However, we prefer to keep addressing the panels by "left", "center", and "right", instead of (a), (b), and (c).

**1.12. Figure 6**

RC: *Figure 6: another very clear figure. However, I wonder why precipitation is presented as mm/6h and the SWC at the panel below in daily values? I guess it is ok if the width of the cell representing a day on the bottom panel is matched with the width 4 bars of rain.*

AR: We think that the vertical bars of 6 hourly rainfall depths provide a helpful supplement to the cumulative precipitation curve, as they display event dynamics more intuitively, while the resolution of 6 hours is a good compromise between hourly and daily. The horizontal width of the bars is matched to the time axis.

**1.13. Table 1**

RC: *Table 1: to address my comment on Figure 1, perhaps you can include an extra column to this table "Dominant soil type" or give percentage of the soil types present within each footprint.*

AR: Please refer to our response to the comment 1.9. We will either include that information in the figure or in the table.

**1.14. Table 2**

RC: *Table 2: Under each header add the units, if applicable. That might avoid having a lengthy caption and faster to understand to readers.*

AR: As suggested, we will include the units in the column headers.

**1.15. Other comments**

**RC:** *Line 9-11: Do you need to enumerate all the static effects and temporally dynamic factors in the abstract? Shorten sentence if possible.*

**AR:** We will shorten the sentence so that it reads: "[...] we apply a state-of-the-art procedure to correct the observed neutron count rates for static effects (heterogeneity in space, e.g. soil organic matter) and for the influence of the dynamic effects (heterogeneity in time, e.g. barometric pressure)."

**RC:** *Line 14: you mention already here "constrained interpolation", which is great. Could you also please mention what you will refer later on as "unconstrained" to improve clarity and keep consistent?*

**AR:** We agree, though we decided to introduce the term "unconstrained" a bit later in the abstract as it provides a better context to understand the difference. The corresponding sentence is then: "The comparison with independent measurements [...] shows that the constrained interpolation approach outperforms the *'unconstrained'* Ordinary Kriging [...]".

**RC:** *Line 44: on "isolated" sensor footprints only. Change "only" for "mainly", as there are already several key studies published using CRNS rovers.*

**AR:** We will replace "only" by "mainly".

**RC:** *Line 51-57: Am I correct that here you present the main aim of the study? If so, please restructure this paragraph first stating the main aim and then the justification. (the content is good, but the flow can be improved).*

**AR:** It is correct that this paragraph outlines the overall aim of the study. We will try to make this clearer in the revised version.

**RC:** *Lines 67 – 78: The first question or specific objective has been given more space compared to questions 2 and 3. Could you please make this section more concise? In that way the "weight" of the three specific objectives seems equally distributed.*

**AR:** We will shorten the description of the first objective in order to achieve a better balance.

**RC:** *Line 94: Abbreviation FDR is not defined previously. Could you please include "Frequency domain reflectometry" here?*

**AR:** We will include "Frequency Domain Reflectometry" to explain FDR at its first occurrence of in the manuscript.

Line 102: change to "We then present and discuss the corresponding results (section 5)..." – as Results and Discussion are presented simultaneously.

**AR:** Will be implemented as suggested.

**RC:** *Line 109: please provide information on what portion of the catchment is covered with grassland (in percent) and forest (in percent) to give a better idea of the heterogeneity in land use to which you refer in Line 60.*

**AR:** We will add the corresponding percentages of forest and grassland coverage.

**RC:** *Line 113: Section 3.1. Overview does not add anything to point 3. Data. In fact, you state "as already pointed out". Thus, I would remove the current Section 3.1.*

**AR:** We will remove section 3.1 of the original manuscript.

**RC:** *Line 138-142: Section 3.3. is unnecessarily long and can be shortened to two sentences. Alternatively fuse current sections 3.3. and 3.6. as one, giving it an appropriate name. The measurements described in these two sections are used to do the continuous correction of the signal.*

**AR:** As suggested, we will shorten section 3.3 (on incoming neutron flux) and merge it with the former section 3.6 (on meteorological data). Altogether, this will become the new section 3.2: "Incoming cosmic-ray neutron flux and meteorological observations".

**RC:** *Line 144: "As we mention in section 1.1. we require local measurements of the soil water content and the soil bulk density" – you don't actually mention soil bulk density in current section 1.1. Either include it there or rephrase this sentence.*

**AR:** We thank the referee for pointing out this inconsistency. We prefer to leave the statement in section 1.1 as it is because the transfer function could also be calibrated for gravimetric soil water content, without the need to determine soil bulk density. Instead, we will adjust the above sentence in the section on "Local measurements of soil water content and other soil variables" simply by dropping the reference to section 1.1 (which is not required as the actual calibration procedure is outlined in the methods section). Hence, the above sentence becomes: "We require local measurements of the soil water content and the soil bulk density [...]"

**RC:** *Line 146: Do "several measurement techniques" refer to the techniques outlines in sections 3.4.1., 3.4.2. and 3.4.3.? If so, please change to "the following measurement techniques".*

**AR:** As suggested, we will change the sentence to "[...] the following measurement techniques were applied to meet these requirements."

**RC:** *Line 168: Current section 3.6 should be moved after current section 3.1., as the meteorological data is relevant to all the CRNS analysis. Also, see my comment on using "climatic gauge", "climate station" or "meteorological station".*

**AR:** Will be implemented accordingly (merged with section on incoming neutron flux).

**RC:** *Line 172: Section 3.7 does not really present data (since you do not use it for analysis and merely to create figure 1). I would remove this section as the information is already provided in the caption of Figure 1. Optionally, add the hyperlink from the current section 3.7 to the caption of Fig. 1.*

**AR:** We agree. Originally, this section was intended to avoid the redundant repetition of the obligatory license acknowledgement. However, we already had to include elongated statements in the figure captions, so we can elongate these just a bit more and drop this section instead.

**RC:** *Line 173: replace "incl." with "including".*

**AR:** Will be done.

**RC:** *Line 411: SMT100 cluster (SoilNet) should be defined/mentioned earlier, as for example in line 90. In line 431 you refer to the same as "FDR-cluster". Choose one term and stick to it.*

**AR:** We prefer to stick with FDR.

**RC:** *Line 484: "… Fig.5 is both encouraging and disappointing" – put first disappointing and then encouraging, as this is the order in which you present them afterwards. Also, why would you say disappointing about a Figure you include in a publication? Choose a different less negative word. There are several places in the manuscript where you downplay the results of your research (see my comment about line 611 for example).*

AR: We will change "disappointing" to "unfavourable" and switch the order to "both unfavourable and encouraging".

RC: *Line 486: "The vertical shades in the left panel indicate that the theta (Ni) is highly uncertain." You refer here to the percentiles, but which ones/which grey shades? The 5-95 percent or the 25-75 percent ones? Be more specific. Also you can see from this left panel that the uncertainty is larger in the wetter range and smaller in the drier range. Acknowledge that instead of saying that they all are highly uncertain. See my comments about downplaying your results.*

AR: We will be more specific with regard to the discussion of the uncertainty as represented by width of the grey shades.

RC: *Line 605-607: While limitations are honestly acknowledged the paragraph ends on a negative note, which does not reflect the main outcome. I would recommend rephrasing along the lines "While both the unconstrained and the constrained approaches failed to fully simulate the dry are in the southwest of the SoilNet area, the constrained captured it relatively better."*

AR: We tend to disagree with that suggestion. This is maybe a misunderstanding, because our sentence "Both models fail to reproduce the dry region in the eastern part of the area" refers to the eastern part, not to the south-western part, as pointed out by the referee. For this eastern dry spot, both interpolation approaches fail the same, so the statement suggested by the referee does not apply. We also do not think that it constitutes a problem that the last sentence of the paragraph addresses an issue that is not a "success". We hence prefer to keep the statement as is.

RC: *Line 611: "largely" outperform seems like an overstatement (downplaying the overall good results of your study). I would remove "largely". If you mean that the unconstrained model outperforms during a prolonged period of time, modify text accordingly.*

AR: We will remove the term "largely", as suggested by the referee.

RC: *Line 622: typo. Change "reliably" to "reliability".*

AR: We will fix the typo.

RC: *Line 629: I understand the willingness to emphasize the "heterogeneous" in here, as it is a key word in this study. However, not entire convinced about "heterogeneous hydro-meteorological conditions". Are you not considering that precipitation, temperature and barometric pressure measured at the climate station are all the same for all sensors? I would recommend saying "dynamic" or rephrase.*

AR: The referee is correct that the term "heterogeneous" might in this context be misleading as it implies heterogeneity in space while we mean, as the referee correctly suspected, heterogeneity in time. We will hence replace "heterogeneous" by "dynamic", as suggested by the referee.

RC: *Line 725: typo. Change "August 2910" to the corresponding year.*

AR: We will fix the typo.

---

## Author Comment (AC2)

**Interactive Discussion: Author Response to Referee #2**

**Spatio-temporal soil moisture retrieval at the catchment-scale using a dense network of cosmic-ray neutron sensors**

Maik Heistermann et al.

*Hydrol. Earth Syst. Sc. Discuss.,* `doi:10.5194/hess-2021-25`
* * *
**RC:** *Reviewer Comment*,    AR: *Author Response*,    ☐ Manuscript text

Dear Referee,

we would like to thank you very much for your positive comments and constructive suggestions to our manuscript. We very much appreciate the time and effort that you have invested in your report.

Please find below our detailed responses to all the issues you have raised in your report. Note that this response addresses both of your reports, part 1 and part 2, in one single document.

We are confident that the manuscript will improve as a consequence to addressing these issues. Yet, the final implementation of changes will also depend on another referee report that is still to be submitted in the interactive discussion.

Kind regards,
Maik Heistermann
(on behalf of the author team)

**2.1. Sensitivity factor**

**RC:** *Sensitivity factor was assumed to be a constant for each sensor, which seems intuitively reasonable but needs scrutiny. Since these factors are essential to the uniform calibration, it at least requires some citations and/or explanation.*

AR: The detector-specific sensitivity or efficiency is a result of manufacturer-level variation of detector gas density, geometrical variation, and configuration parameters related to electricity. All of those were fixed once during manufacturing and cannot change over time. [Schrön et al., 2018] have shown that the resulting count rate efficiency is different from sensor to sensor, while significant variation in time is not evident. This is also known from experience with long running sensors of the COSMOS network in periods of more than a decade, where the duration of our 2-month campaign is negligible. On the basis of these explanations, we think that the application of constant, sensor-specific efficiency values is sufficient.

We will add a brief corresponding statement and citation in section 4.1.1 of the manuscript ( Standardization of sensitivity).

**2.2. Local uncertainty**

**RC:** *Please define/specify "local uncertainty". I think the "local" refers to the parameter space, not the spatial-temporal space. If I am correct, it creates some ambiguity in the text since the discussions are always related to space/time in this paper. (Line 271, 415, 665, etc.)*

**AR:** We apologize for the ambiguity, and we fully agree that the term "local uncertainty" requires a clear definition. In fact, we refer to "local uncertainty" as the uncertainty of our soil moisture estimate for a specific sensor footprint $i$, i.e. as the uncertainty of $\theta(N_i)$ and $\theta_i^{obs}$, expressed as the width of the interval between two quantiles. It should also be clear that this uncertainty only refers to the point in time at which the manual soil moisture measurement campaign took place which was the basis of the $N_0$ calibration.

In the revised version of the manuscript, we will explain the term "local uncertainty" in section 5.2 where the actual results are discussed. In section 4.3, l. 271 of the original manuscript, we will avoid the use of the term instead of already defining it. That is because we think that the meaning of the term becomes more tangible for the reader in the context of the presentation of the actual results, hence we should not introduce it before.

**2.3. Monte-Carlo simulation**

**RC:** *Please briefly explain the choice of "200 times" of the Monte-Carlo simulation on the sensitivity of N0. To my understanding, the number of simulations depends on the dimensions of the parameter space. Why are 200 times good enough to quantify the uncertainty of $N_0$ concerning these many parameters and disturbances.*

**AR:** The referee is correct that it would be good to have a formal justification of the number of runs that constitute our Monte-Carlo analysis. However, the parameters, their assumed probability distributions, and the corresponding stochastic disturbances are very different from each other. That makes it difficult to apply formal frameworks to assess the required numbers of runs. For example, some disturbances are rather a sub-sampling (e.g. the determination of the time interval over which the neutron count rates are averaged, or the selection of soil profiles that are included in the interpolation); for other input parameters, the definition of the underlying distribution and its parameters is necessarily arbitrary (e.g. the Kriging range, or the water equivalent from soil organic carbon and lattice water). Given these difficulties, we have addressed the issue rather pragmatically: we found that the results of the Monte-Carlo simulations with using 200 runs are robust, meaning that they do not vary substantially from simulation to simulation with regard to the output we were looking at (which is specifically the interquartile range, while the range between the 5th to the 95th percentile is purely for illustrative purposes). We also found that the results did not substantially change when we increased the number of runs per Monte-Carlo simulation.

We would like to emphasize that the Monte-Carlo-analysis is, in the context of this study, of rather qualitative relevance: its main purpose is to demonstrate that the disagreements that we observe in Fig. 4 can mostly be explained by the local uncertainties of $\theta(N_i)$ and $\theta_i^{obs}$, while location 7 is clearly different.

In our view, arbitrary decisions in the design of the Monte-Carlo cannot be avoided at this point, and we have also been open with that in the conclusions. Yet, we fully understand and appreciate the referee's concern in this context. As a response, we suggest to very briefly mention, in section 4.3, the level of arbitrariness involved in this analysis, and the corresponding limitations in the interpretation of the results. Still, we could further increase the number of Monte Carlo runs if desired.

**2.4. 2.4 Models**

**RC:** *Line 88 References for the concept of geophysical inversion are needed.*

AR: We suggest to cite [Zhdanov, 2015] in this context as a reference to the fundamental idea of geophysical inversion.

RC: ***Line 329 There are three parameters for a variogram model, nugget, sill, and range. The paper only emphasized the range but did not mention the other two. Please specify the parameter selections.***

AR: We apologize for the incomplete documentation. We did not specify nugget and sill, as these do not affect the result of the predicted variable, but only the Kriging variance. Since we do not use the latter, nugget and sill can be chosen arbitrarily (in our case: nugget=0, sill=1). Nugget and sill become important when a theoretical variogram model is fitted to an empirical semi-variogram, as the choice of nugget and sill might affect the range, when the three parameters are fitted together. In our case, we did not fit a variogram model. Instead, the choice of the range of 300 m was rather a preference to express the scale at which we are interested in representing soil moisture heterogeneity.

Altogether, we will clarify these aspects in the revised version of the manuscript, and also state the values of nugget and sill used for our calculations.

RC: ***Line 297 The Kriging ranges for soil moisture and bulk density are quite different. Please justify this selection.***

AR: We agree that this should be explained better. The sampling intervals for the Kriging ranges in the Monte Carlo analysis (section 4.3) were based on the Kriging range values used for the interpolation of the soil variables as outlined in section 4.1.4, which were 50 m for all soil variables except soil moisture (150 m). These values were not obtained from fitting a variogram model, but rather heuristically: we chose a higher range value for soil moisture because the resulting estimates of $\theta^{obs}$ were more consistent with $\theta(N)$, although a systematic optimisation was not carried out. We addressed the apparent arbitrariness of this procedure by defining a sufficiently large interval around these range values from which we would sample in the Monte-Carlo analysis. We will point out, in the revised manuscript, that the selection of the Kriging range values could, in future studies, be subject to further systematic optimisation.

**2.5. Footprint, model parameters, and scaling**

RC: ***One of the unique features of CRNS is its large footprint, which could directly influence data visualization, model selection, and interpolation. The grid size for the interpolation process is 10 m * 10 m (Line 311), which is much smaller than the footprint. This implies that the modeling is not just an interpolation but also involves a downscaling process for the CRNS measurements. It is of great interest in terms of the CRNS studies. However, it also requires more clarification and cautiousness. For example, is it reasonable to use observed soil moisture, $\theta(N_{obs})$, to do Ordinary Kriging with a resolution much smaller than its footprint? Does it implicitly assume that observed soil moisture values are also representative at a smaller scale?***

AR: We thank the referee for this comment. Obviously, he or she is entirely right to demand that cautiousness. We hoped to express that caution with our statement from ll. 311-314 of the original manuscript:

> [ll. 311-314] The grid resolution is arbitrarily selected, and does not necessarily reflect the resolution at which the grid effectively conveys information of spatial heterogeneity; in other words, the product should not be interpreted at the scale of 10 m. Still, we require this comparatively fine horizontal resolution since some of the following steps require to re-aggregate (i.e. to average) the spatial soil moisture estimates inside a CRNS footprint.

Accordingly, we do not actually aim to represent soil moisture variation between 10 m grid tiles, but we require that resolution in order to reasonably apply the forward operator in order to obtain neutron intensity from a spatial soil moisture grid. In addition, one could see this sub-footprint resolution as a tool to represent gradients in the footprint, rather than values of the single cells. In the revised manuscript, we will attempt to clarify this more.

Furthermore, we emphasize in ll. 328-332 that Kriging is used as a "model" to represent our notion how soil moisture varies at a specific scale:

> [ll. 328-332] In this study, let us assume that the spatial distribution of soil moisture in the study area is smooth and continuous, and that this spatial pattern could be represented by a model $m$ that corresponds to Ordinary Kriging with an exponential variogram model and a range parameter of, say, 300 m, using the CRNS sensor locations as points of support. We hope it is clear to the reader that the choice of such a model is arbitrary and subjective, although it should be based on our "expert" notion of how soil moisture varies at a specific scale.

Again, we will attempt, in the revised version, to emphasize that the use of Kriging with a grid resolution of 10 m does not mean that we should interpret variability at that scale.

Finally, the referee wonders whether we "implicitly assume that $[\theta(N)$ is] representative at a smaller scale". Our answer would be no, although it is true that the unconstrained model, technically, reproduces $\theta(N_i)$ at the sensor location $i$. However, that is rather a side effect and not a necessary requirement. The key property of our model $m$ is that it represents soil moisture variation at a scale that is given by our (arbitrary) choice of the variogram (exponential with a range of 300 m). Please note that other models might well be able to represent soil moisture heterogeneity at an even finer effective resolution. Such a model could be a statistical relationship between surface properties (soil, terrain, vegetation) and soil moisture, or a physically-based model (see ll. 319-324). The effective resolution would be subject to their validity at a finer scale as well as the accuracy of their input data (please also refer to our response to next comment).

> [ll. 319-324] What we refer to as the "unconstrained" approach could imply any kind of (geostatistical) model or assumption $m$ that represents the spatial distribution of soil moisture, $\theta$, on the basis of any parameter set $\mathbf{p}$. For example, $m(\mathbf{p})$ could be the nearest neighbour algorithm. In that case, $\mathbf{p}$ would be the soil moisture values at a set of sampling points. As another example, $m(\mathbf{p})$ could be a statistical relationship between landscape attributes and soil moisture, hence $\mathbf{p}$ would comprise the parameters of that statistical model. Or, $m(\mathbf{p})$ could be a physical model of water movement in soils, with $\mathbf{p}$ being the entirety of (potentially spatially distributed) model parameters.

**RC:** *The design of the forward operator and the optimization argument is innovative since it provides a way of downscaling CRNS measurement to almost any arbitrary scale/resolution, which may be only limited by computational capacity.*

*The design of the dense network made the footprints of CRNS largely overlapped, which provides extra information about soil moisture spatial patterns. This may also make it logically possible and reasonable to do the downscaling and to improve the interpolation. Can the overlaps be used for results validation?*

AR: We agree, in general, with the referee's view that the use of the forward operator allows for a certain level of downscaling (see our response to comment 2.5), and it certainly is one of the specific aims of this study

to demonstrate that potential. However, we do not think that the achievable resolution is purely a matter of computational resources. In our view, it is rather a matter of how well our model $m$ is able to represent patterns at high resolution. Example: We could enhance, in our setup, the spatial resolution of our target grid from 10 m to let's say 10 cm. That would involve a substantial increase in computational costs for the interpolation and the application of the forward operator, yet the effective/meaningful resolution of the results will not be higher than before.

Somewhat related to that point is the aspect of overlap: in general, we would expect that a large overlap from multiple sensors would help to better constrain the inverse problem, yet it does not, in our view, provide "extra information" for an independent validation. Even with a strong overlap, we still need a model of spatial soil moisture variation to make the problem solvable. The advantage of the overlap is particularly that the parameters of that model will probably be constrained better because changes of soil moisture in the region of overlap will affect multiple footprints and hence multiple values of $N^{sim}$.

While this discussion is certainly interesting, we would prefer not to extend it further in the context of this manuscript. We see the present study as a proof-of-concept, and both practical and theoretical aspects should be explored in future studies, as also outlined in ll. 722-730 of the original manuscript.

**2.6. Technical Comments**

**RC:** *Line 26 "small spatial measurement support" and Line 330 "points of support". Support is an important concept in defining spatial scales of soil sampling and measurements. I recommend adding a definition and citations here. This would also help to present the results on soil moisture spatial patterns in the following sections.*

AR: In the revised version of the manuscript, we will refer, in section 1.1., to [Blöschl and Grayson, 2000] as the key reference with regard to the concept of spatial support in the observation and interpolation of spatial variables. We will also better explain, around l. 330 of the original manuscript, the meaning of "points of support", as this term does not refer to the concept of "measurement support" in the sense of [Blöschl and Grayson, 2000], but to the "nodes" of the interpolation, i.e. the locations at which an observation is assumed to be available. Alternatively, we could replace "points of support" by "node".

**RC:** *Line 259 "assuming a spatially uniform value of $N_0$..." Modification required. Since $N_0$ mainly depends on the sensor itself after correcting all factors (air pressure, vegetation, lattice water, etc.), it is not a spatial variable.*

AR: We agree that this is misleading. We dropped "spatially" so statement becomes "[...] assuming a uniform value of $N_0$ [...]".

**RC:** *Line 262 Eq. 1 - recommendation: replace comma with semicolon, i.e. $\theta(N_i; N_0)$. To my understanding, $N_i$ is a variable, and $N_0$ is a parameter in Eq. 1.*

AR: We thank the referee for the suggestion, but we would prefer to keep the notation as it is: while in the context of Eq. 2, $N_0$ is the parameter (which is optimized), the constrained interpolation treats all $N_i$ as parameters.

**RC:** *Line 350 delete extra "suitable"*

AR: Thanks, will be deleted.

**RC:** *Line 622 reliably -> reliability*

AR: Will be corrected.

**References**

[Blöschl and Grayson, 2000] Blöschl, G. and Grayson, R. (2000). Spatial observations and interpolation. In Blöschl, G. and Grayson, R., editors, *Spatial Patterns in Catchment Hydrology - Observations and Modelling*, chapter 2, pages 17–50. Cambridge University Press, Cambridge.

[Schrön et al., 2018] Schrön, M., Zacharias, S., Womack, G., Köhli, M., Desilets, D., Oswald, S. E., Bumberger, J., Mollenhauer, H., Kögler, S., Remmler, P., Kasner, M., Denk, A., and Dietrich, P. (2018). Intercomparison of cosmic-ray neutron sensors and water balance monitoring in an urban environment. *Geoscientific Instrumentation, Methods and Data Systems*, 7(1):83–99.

[Zhdanov, 2015] Zhdanov, M. S. (2015). Chapter 1 - forward and inverse problems in science and engineering. In Zhdanov, M. S., editor, *Inverse Theory and Applications in Geophysics (Second Edition)*, pages 3–31. Elsevier, Oxford, second edition edition.

---

## Author Comment (AC4)

**Interactive Discussion: Author Response to Referee #3**

**Spatio-temporal soil moisture retrieval at the catchment-scale using a dense network of cosmic-ray neutron sensors**

Maik Heistermann et al.
*Hydrol. Earth Syst. Sc. Discuss.,* `doi:10.5194/hess-2021-25`
* * *
**RC:** *Reviewer Comment*,    AR: *Author Response*,    ☐ Manuscript text

Dear Referee,

we would like to thank you for your critical comments and constructive suggestions to our manuscript. We very much appreciate the time and effort that you have invested in your report.

Please find below our detailed responses to the issues you have raised in your report. We are confident that the manuscript will improve as a consequence to addressing your concerns.

Kind regards,
Maik Heistermann
(on behalf of the author team)

**3.1.  Advancement of practical applications**

**RC:** *This manuscript describes the first ever attempt to use multiple CRNS to estimate catchment-scale soil moisture, including temporal and spatial variations. The research conducted is of interest and importance to the research community, even if the findings are somewhat marginal in their advancement of practical application.*

AR:   The campaign itself and the present analysis were designed to answer primarily theoretical and methodological questions of cosmic-ray neutron sensing, and not geared towards practical applications. We agree that it will, for the foreseeable future, not be cost-effective or practical to deploy 10-20 CRNS probes per square kilometer for routine soil moisture monitoring (see also our response to comment 3.9). And yes, the results of the present analysis, in terms of the soil moisture patterns retrieved from our CRNS network, might not appear overwhelming, as opposed to the spatial detail that could be achieved with wireless sensor networks (see also our response to comment 3.8).

As the referee pointed out correctly, this analysis is the first attempt to use the observations of such a dense network for a spatially explicit retrieval of soil moisture patterns, and it has a clear methodological focus: (1) we demonstrate a uniform homogenisation and calibration approach across multiple CRNS stations (a concept that will be relevant particularly for future CRNS observatory networks), and (2) on that basis, the potential to capture main features of spatial soil moisture variability at the scale of hectometers, continuous in time. We also introduce an entirely new approach to constrain a spatial interpolation by the dynamic CRNS footprint

characteristics. What this study does *not* claim to achieve: to provide a scalable monitoring technique, to represent small-scale features of soil moisture variability, and to consistently link our observations to hydrological and soil hydraulic processes (although we briefly discuss potential explanations of temporal dynamics in section 5.3 of the original manuscript).

While this study does explicitly not claim to *achieve* these objectives, it is still *motivated* by them and works towards those goals. For example, future research should aim to combine such CRNS observations with spaceborne remote sensing data: to obtain patterns at a higher resolution via downscaling, or to upscale CRNS observations beyond such local networks in a cost-efficient way. And we should design assimilation experiments with hydrological models at various levels of physical detail in order to understand how we can consistently represent our observations and our physical expectations. We do not solve these issues in the present study, but we outline a methodological framework that should facilitate them.

We apologize if the scope of the present manuscript has not come across clear enough. In the revised version, we will attempt to clarify the scope, but also to better outline prospective research threads that will hopefully unfold a more visible impact.

**3.2. Length of the manuscript**

RC: *While the paper is well-organized and written, it is excessively long and tedious to read. I understand that the authors wish to share the minutia of their novel methodology, but as written the paper is difficult to read and seems more akin to a grant proposal than a scientific manuscript.*

AR: We thank the referee for this comment. It is in line with the comments of referee #1 who also suggested to shorten the manuscript. Then again, referee #2 rather required *more* details in terms of both methodology and discussion. In summary, we agree that there is potential to make the paper more concise, and we have outlined, in reply to the very constructive suggestions of referee #1, how to achieve that goal. At the same time, we tend to insist that due to the strong methodological focus of the paper, the adequate documentation of methodological details and the discussion of their limitations should not suffer. We are confident, though, to find a balance between these two ends in the revised manuscript.

**3.3. Absolute values**

RC: *I dislike the goal of the project to match the pattern rather than absolute values (stated in Line 434) and the general avoidance of quantifying differences or variability in the estimated and measured soil moisture values. I understand why the authors have chosen this approach, but I also think that the use of the absolute values of measured and estimated volumetric water content would be useful to present to the scientific community. Relative values only provide so much information.*

AR: There is maybe a misconception of the study, as it has not been "the goal of the project to match the pattern rather than absolute values", and from our perspective there is no "general avoidance of quantifying differences or variability" in the manuscript.

The reviewer refers to line 434 of our manuscript:

> [...] to eliminate the potential effect of systematic bias in the SoilNet data, [...] we target the matching of the pattern rather than the absolute values."

This statement only applies to the comparison to the SoilNet data, and explains why we use Spearman's rank

correlation coefficient as an objective measure of similarity between the soil moisture inferred from the SoilNet and the interpolation of CRNS-based soil moisture estimates: in this case, we are specifically interested in how well the spatial patterns agree, in order to evaluate which interpolation method better represents soil moisture gradients at the hectometer scale. Numerous equations exist for converting permittivity (the prime observational variable of the SoilNet) to volumetric soil moisture [Mohamed and Paleologos, 2018], and the exact shape could differ substantially, depending on underlying functions and corresponding coefficients. To eliminate the associated uncertainty and arbitrariness from our analysis, we chose to use Spearman rank correlation.

But we agree that the above formulation is ambiguous, so we suggest to change the paragraph to:

> We evaluate the similarity of the spatial soil moisture patterns obtained from the FDR-cluster and the interpolation of $\theta(N_i)$ for each day from May 20 to July 15, 2019. As a measure of similarity, we chose Spearman's rank correlation of the corresponding soil moisture grids. Using that measure, we eliminate potential effects of uncertainty in the soil moisture values obtained from the SoilNet, as the conversion from permittivity to volumetric soil moisture can be subject to systematic bias [Mohamed and Paleologos, 2018].

While we are convinced that this comparison approach is adequate in the context of this study, we would still like to address the referee's concern of a "general avoidance of quantifying differences or variability in the estimated and measured soil moisture values." To that end, we would like to adopt the referee's suggestion made in comment 3.8: *"[...] I would be more interested in a time series figure showing the mean field-scale volumetric water content for each of the three scenarios- unconstrained, constrained, and SoilNet. I suspect that the constrained and unconstrained values would be far more similar to one another than to the SoilNet values."* Certainly, the referee is correct with assuming that the two CRNS-based interpolation results (constrained, unconstrained) are far more similar to each other than each of them is to the SoilNet estimates. We think that this notion is obvious from the soil moisture maps shown in Fig. 9. Still, we would like to meet the referee's demand, and suggest to add another panel to Fig. 10 in which we show the soil moisture time series for the three scenarios for the area of the SoilNet. We interpret the term "mean field-scale volumetric water content", as suggested by the referee, as the mean over the SoilNet area.

**3.4. Shallow groundwater**

**RC:** *Section 2: The authors mention in passing the presence of very shallow groundwater, but do not mention how this likely has significant effects on their CRNS measurements. This issue should be addressed much more thoroughly.*

**AR:** We agree that the interpretation of soil moisture patterns would benefit from a better knowledge of the depth to the groundwater table. Unfortunately, we rely on the references given in section 2 of the original manuscript. Spatially explicit information on depth to the groundwater table are not available, and we assume that the below-ground structure is complex and heterogeneous: the very shallow groundwater, where present, appears to result from the local accumulation of percolating water on low-permeable soil layers – a perched aquifer, if at all. As such, it would not be a laterally extended water body that feeds water to the top-soil but rather a part of the soil water balance. All this information, however, is patchy, and rather hypothetical, and does not rely on robust spatial observations of shallow groundwater. Hence, we would prefer not to excessively hypothesize about this matter in the manuscript. However, we suggest to mention, in the revised manuscript, the possibility of such local structures, but also emphasize the lack of corresponding data.

**3.5. Soil texture and SOM**

**RC:** *Additionally, the authors should provide the textural and SOM information for samples taken near each CRNS (texture) and for each mixed sample (SOM). Providing an average catchment-scale value is not acceptable, and readers cannot be expected to read every cited paper to find this information, which could easily be incorporated into Table 2.*

**AR:** We agree that it would be helpful to explicitly link soil attributes to CRNS locations in order to allow for a better interpretability. A soil texture analysis was unfortunately not carried out for the soil that was sampled by cylinders near each CRNS. Based on the referee's request we have, however, computed the relative coverage of soil types as extracted from the Übersichtsbodenkarte (1:25000) of the Bayrisches Landesamt für Umwelt [Bayerisches Landesamt für Umwelt, 2014] and soil texture classes as obtained from the Bayrisches Landesamt für Digitalisierung, Breitband und Vermessung within the product Bodenschätzungsdaten [Landesamt für Digitalisierung, Breitband und Vermessung, 2018], using the horizontal weighting scheme from [Schrön et al., 2017]. The soil organic matter (SOM) content was only determined for mixed samples from the three landuse/soil classes (forest/mineral, grassland/mineral, and grassland/organic). In order to meet the referee's demand, we suggest to include the SOM content (weighted average in the footprint) as an additional attribute in Tab. 2. As for the soil texture class and type, we suggest to include the dominant soil texture class and soil type in Tab. 1 instead of Tab. 2, as the dominant land use type in the footprint is also provided in that table.

**RC:** *Further, the estimated gravimetric water content values > 1.0 g/g shown in Figure 4b are unrealistic, unless you are considering a highly organic soil. However, based on the information presented, it is impossible to determine the soil type near CRNS 21 and 23.*

**AR:** The referee is correct in noting that the high gravimetric soil water content values for the CRNS sensors 21 and 23 are due to the fact that the footprints of these sensors are dominated by highly organic soils. With the additions to Tab. 2 as suggested above, this should become more transparent to the reader – so we thank the referee for this suggestion.

**RC:** *Table 2: In addition to including information regarding soil texture and SOM for each site, the authors should include some indication of the variability of the values shown for each site.*

**AR:** Quantifying the variability of soil texture and SOM in the sensor footprint is difficult due to the limited amount of data (as pointed out above): high-resolution soil mapping was, unfortunately, not part of the underlying campaign. Hence, we suggest to stick with the additional information in Tab. 2 as outlined in our response above.

**RC:** *Also, bulk density values are incredibly low, less than 1 kg/L in most cases. Including the SOM content in this table would make those values look less suspect, if indeed the SOM content is extremely high. If not, the authors need to address the very low bulk density values reported.*

**AR:** The bulk density values shown in Tab. 2 are in fact rather low. It is important to consider, though, that these bulk density values represent the weighted average bulk density per sensor footprint, with the weights corresponding to the vertical and horizontal sensitivity as given by [Schrön et al., 2017]. The low bulk density values here are hence, partly, a result of the vertical weighting: the bulk density generally *decreases* towards the surface (as can be seen in the vertical profiles shown on the supplementary, Fig. S2), while sensor sensitivity pattern *increases* towards the surface. So while the lowest bulk density values are in fact due to organic, high porosity soils (sensors 21 and 23), the generally low bulk density values in Tab. 2 are also due to the fact that the vertical weighting emphasizes the less-dense top soil. We thank the referee for pointing out this issue as we think it would merit a brief clarification in the revised version of our manuscript.

**3.6. Uncertainty of $\theta^{obs}$**

**RC:** *Figure 5: How does the variability in $\theta_i^{obs}$ during Monte Carlo simulations compare to the variability observed in measured volumetric water content from thermo-gravimetric samples?*

**AR:** $\theta_i^{obs}$ or its variability in the Monte Carlo analysis should not be compared to the volumetric soil moisture content as obtained from the thermo-gravimetric samples. As pointed out in section 3.4.2 of the original manuscript, we only have *one* thermo-gravimetric sample close to each CRNS sensor. $\theta_i^{obs}$, in turn, is the result of an interpolation of thermo-gravimetric samples *and* FDR-measurements, and subsequent vertically and horizontally weighted averaging. Hence, we do not see the reason for such a comparison.

**RC:** *Could the authors provide mean uncertainty values for Monte Carlo $\theta(N_i)$ and $\theta_i^{obs}$? It is difficult to estimate these values from Figure 5 alone.*

**AR:** We are not sure what the referee means by *"mean uncertainty values for Monte Carlo $\theta(N_i)$ and $\theta_i^{obs}$"*. We would like to emphasize that Fig. 5 only represents the uncertainty of footprint-scale soil moisture estimates at the time of the calibration (June 25-26, 2019), based on the Monte Carlo analysis. The uncertainty is, in Fig. 5, shown as the inter-quartile range of the soil moisture estimates, as well as the range between the 5th and 95th percentile. Does the referee refer by "mean uncertainty" to the average of these percentiles over all 18 CRNS footprint represented in Fig. 5? If yes, we do not see what could be learned from such an average. As already pointed out to referee #2, the main motivation of Fig. 5 is to visually demonstrate that the disagreements that we observe in Fig. 4 can mostly be explained by the local uncertainties of $\theta(N_i)$ and $\theta_i^{obs}$, while location 7 is clearly different. We would hence prefer not to provide average uncertainty values, if the referee agrees.

**RC:** *Line 543: It would be good, again, if the authors would quantify this uncertainty.*

**AR:** We would like to refer to our response above: as the local uncertainty, i.e. the uncertainty of our soil moisture estimate for a specific sensor footprint, is very variable across sensor locations, we do not see the added values in providing an average value. We hope that the referee will agree.

**3.7. Reference evapotranspiration**

**RC:** *Line 525 and Figure 6a: Reference evapotranspiration is different than potential evapotranspiration, but the terms are used interchangeably in the text. Make sure all instances are changed to "reference."*

**AR:** We will change all instances to "reference evapotranspiration".

**3.8. Comparison to SoilNet**

**RC:** *Figure 9. The results shown in this figure are underwhelming. I expected that the highly concentrated CRNS sensors would be able to provide a closer match in spatial soil moisture distribution to the Soil-Net measurements. The lack of spatial agreement with SoilNet soil moisture values, even with a high concentration of CRNS that is unlikely to replicated in practice, is surprising and a bit disappointing. If the CRNS are unable to provide useful spatially explicit information, what is the benefit of using these extremely expensive sensors rather than many cheaper in-situ sensors or downscaled remote sensing data? Also, I would be more interested in a time series figure showing the mean field-scale volumetric water content for each of the three scenarios- unconstrained, constrained, and SoilNet. I suspect that the constrained and unconstrained values would be far more similar to one another than to the SoilNet values.*

AR:    We have already responded to parts of this comment in our response to comments 3.1 and 3.3, so we would like to keep this response brief. We understand that the visual impression from Fig. 9 might be disappointing at first - but only if you expected to reproduce the meter-scale variability as represented by 300 invasive sensors (SoilNet, 50 profiles, 3 depths, 2 redundant measurements per depth) by 4 (inside) or 6 (near SoilNet area) non-invasive hectare-scale CRNS sensors. We hope that the referee does not misinterpret our goal as "advertising" dense CRNS networks as "the" solution (according to the comments of referee #1, we are excessively critical to our results rather than sugarcoating them). As matching meter-scale variability cannot be the goal of a study using just a few hectometer-scale CRNS stations, we would like to clarify again: Our main finding is that dense CRNS networks can help to represent soil moisture gradients at the hectometer scale, in an area of say $1\,km^2$, and that the inversion-style technique of constrained interpolation can add to that. In combination with auxiliary information such as remote sensing, we might even be able to add further details to such patterns - corresponding studies are underway.

Please allow a short remark regarding the mentioned proportionality, practicability, and costs: a high number of in-situ sensors such as the WSN/SoilNet can also become very expensive in terms of long-term maintenance and disruptive in terms of installation, which are clear limitations especially for intensely managed agricultural fields. In contrast, CRNS sensors require hardly any maintenance and sense soil moisture non-invasively at a larger scale, while they can be used in small networks or mobile roving campaigns. We think that these advantages could be well suited for agricultural applications as well as evaluation of remote-sensing and modeling products, where intermediate-scale soil moisture patterns are of key importance.

**3.9.  Costs**

RC:    *Line 595: While I agree that the current application of a large number of CRNS in a small area is interesting, I do not think it is economically feasible or sustainable. The authors should at least mention the cost prohibitive nature of this study.*

AR:    As already pointed out in our response to comment 3.1, we agree that such dense networks are not suited for routine soil moisture monitoring, and while we have not suggested otherwise in our manuscript, we will explicitly clarify that in the revised version.

In research environments, we hope that such dense networks can contribute to a better understanding of soil moisture variability in space and time, and that other sensor platforms, such as spaceborne remote sensing, in combination with hydrological models, could be used for a cost-efficient upscaling of these insights.

**3.10.  Technical corrections**

RC:    *Line 301: Should read "how the availability of sampling locations affected N0 calibrations [...]*

AR:    Thanks for pointing out this incomplete sentence, we changed it accordingly.

RC:    *Line 725: Should read "from August 2019"*

AR:    The typo was corrected, thanks for pointing it out.

**References**

[Bayerisches Landesamt für Umwelt, 2014] Bayerisches Landesamt für Umwelt (2014).  Übersichtsbo-denkarte tk25-blatt 8132. www.lfu.bayern.de.

[Landesamt für Digitalisierung, Breitband und Vermessung, 2018] Landesamt für Digitalisierung, Breitband und Vermessung (2018). Bodenschätzung. `https://geoservices.bayern.de/wms/v1/ogc_alkis_bosch.cgi?`

[Mohamed and Paleologos, 2018] Mohamed, A.-M. O. and Paleologos, E. K. (2018). Chapter 16 - dielectric permittivity and moisture content. In Mohamed, A.-M. O. and Paleologos, E. K., editors, *Fundamentals of Geoenvironmental Engineering*, pages 581–637. Butterworth-Heinemann.

[Schrön et al., 2017] Schrön, M., Köhli, M., Scheiffele, L., Iwema, J., Bogena, H. R., Lv, L., Martini, E., Baroni, G., Rosolem, R., Weimar, J., Mai, J., Cuntz, M., Rebmann, C., Oswald, S. E., Dietrich, P., Schmidt, U., and Zacharias, S. (2017). Improving calibration and validation of cosmic-ray neutron sensors in the light of spatial sensitivity. *Hydrology and Earth System Sciences*, 21(10):5009–5030.

---

## Author Response (AR1)

**Author reponse to referee comments**

**Spatio-temporal soil moisture retrieval at the catchment-scale using a dense network of cosmic-ray neutron sensors**

Maik Heistermann et al.

*Hydrol. Earth Syst. Sc. Discuss.,* `doi:10.5194/hess-2021-25`
* * *
**RC/EC:** *Reviewer/Editor Comment*,     AR: *Author Response*,     ☐ Manuscript text

Dear Referees, dear Editor,

we would like to thank you again for your comments and constructive suggestions to our manuscript. This letter contains our final responses which also formed the basis for the revision of the manuscript. In most parts, these responses correspond to the ones we had already given in the interactive discussion. In some cases, we also found a better way to address the requirements, and outlined the approach accordingly in our reponse. We have also added a response to the specific comments of the Editor.

Several comments, specifically from referee #1 and #3, as well as the Editor, required shortening the manuscript. At the same time, referees #2 and #3 as well as the Editor required additional details with regard to methods, results, and discussion. In total, the revised manuscript is now five pages shorter, and we hope to have found an adequate balance between both requirements.

In summary, the revised version of the manuscript has substantially changed and, in our view, improved. We hope that this revised version is now acceptable for publication in HESS.

Kind regards,
Maik Heistermann
(on behalf of the author team)

**1. Response to referee #1**

**1.1. General comments**

**RC:** *[...] my main concerns are with the presentation quality of the paper: the length and structure of the manuscript could be substantially improved (see specific comments). Therefore, I would like to suggest major revisions.*

AR:  We have restructured and shortened the manuscript according to the following suggestions of the referee (see our responses below).

**1.2. Structure**

**RC:** *While this is a discussion paper, I would encourage the authors to be more concise and structured when presenting their findings (examples are given below). There are sections which seem to be excessively long. That would hopefully help address another issue, which is the often "downplaying" of the study findings (examples also given).*

AR: We shortened the manuscript where possible, specifically section 6 ("Summary and conclusions"). We also appreciate the referee's concern on "downplaying" the results. We removed statements that appear unnecessarily critical towards the outcomes of our own study. Yet, we consider it important to clearly and unambiguously emphasize any limitations of our results, especially as this is the first attempt of inferring spatio-temporal representations of soil moisture from a dense CRNS network, and as we had to make a number of assumptions. This was also required by referees #2 and #3. We hope that we found an adequate balance between these two requirements.

**RC:** *Moreover, during the discussion the implications of the findings and potential benefit of the application of these novel analyses are shown/discussed but in a scattered manner throughout "Results and discussion" section. I would recommend condensing them in a subsection within "Summary and conclusions". Below I outline specific examples:*

AR: We substantially revised section 6 (summary and conclusions): besides a very short summary, it now focuses on highlighting main lessons as well as theoretical and practical implications for future research. That way, it has become more concise, but also puts more emphasis on the implications. We hope that this is in line with the referee's requirement.

**1.3. Presentation of aims and objectives**

**RC:** *In Section 1.2. Aims and objectives: research questions and specific objectives of the study are presented between lines 65-95 and surely could be condensed. If you would wish to keep all the text, separate clearly in "aims and objectives" and "justification". That would help structuring further the manuscript.*

AR: We agree that it would improve the readability if we moved the specific objectives of the study to a dedicated subsection (1.3: Specific objectives). We also moved the overview of the manuscript structure (ll. 96-102 of the original manuscript) to a separate subsection (1.4: Manuscript structure).

**1.4. Section 4.4**

**RC:** *Section 4.4. has effectively three subsections "Unconstrained model", "Constrained model" and "Forward operator". Please give these index numbers 4.4.1., 4.4.2 and 4.4.3. (similar to what you have done in section 4.1.). Also, lines 319-340 from subsection Unconstrained model: please try to shorten and restructure. Only in line 327 you say what you have actually done in the study and before that you give several examples (319-327). I would encourage you to first say what you did and then give the examples.*

AR: As suggested by the referee, we numbered the subsections "Unconstrained model", "Constrained model" and "Forward operator". We also followed the suggestion to shorten and reorder the description of the unconstrained model. After a brief introduction of the idea of a parametric model $m(\mathbf{p})$, we present the actual model used in this study, and, on that basis, explain why it is "unconstrained".

**1.5. Section 5.2.**

**RC:** *Section 5.2. between lines 473-484. – text can be shortened, and arguments presented more concisely. For example, line 472 "Clearly, the agreement is less than perfect. Still, the general pattern suggests", can be rephrased to "While the agreement is not perfect, the general pattern suggests...".*

**AR:** We adopted the referee's suggestion for the first sentence of that paragraph, and further shortened the text that originally extended from ll. 473-484.

**1.6. Align section 5 with specific objectives**

**RC:** *In general, in Section 5: try to restructure main discussion points (5.1. to 5.5.) in a way that it follows the flow of your aims and objectives.*

**AR:** We agree that it would be helpful to better align the structure of the results section with the study objectives/research questions. To that end, we applied two major changes to section 5:

First, we moved the former section 5.1 ("How robust is the standardization of the sensitivity?") to the supplementary section S2. While the former section 5.1 is an important foundation of our study, it was not a major research subject and has, to some extent, already been addressed in [Fersch et al., 2020]. We think that the details are worth being reported in the supplementary (specifically the comparison to the roving data), while the key result - the successful standardization - is just very briefly stated in section 5.1 of the revised manuscript (which is now "Can we find a uniform $N_0$ for the entire study area?").

Second, the former sections 5.4 and 5.5 were merged into a single section ("5.3 How does soil moisture vary within and between the sensor footprints?") with two sub-sections ("5.3.1 Spatial interpolation of CRNS-based soil moisture" and "5.3.2 Comparison of spatial patterns against SoilNet observations").

As a result of these changes, the research questions 1-3 (as pointed out in section 1.3 of the revised manuscript, see comment 1.3) now exactly correspond to the sub-sections 5.1 to 5.3 in the "Results and Discussion" section of the revised manuscript.

**1.7. Summary and conclusions**

**RC:** *Section 6 Summary and conclusions (which are then called "main lessons learned along these steps") needs to be more concise. Shortly introduce the "main lessons" and then expand on them in 6.1. and 6.2.*

**AR:** Please refer to our response to comment 1.2. We substantially shortened the section and put more emphasis on implications for future research.

**1.8. Practical guidance for future dense CRNS networks**

**RC:** *Section 4.3. is particularly good, as uncertainties have been very well acknowledged and explained. Moreover, the limitations of the study are thoroughly and openly discussed. However, I encourage the authors to include a short comment in discussion on the applicability and reproducibility of the set-up (of dense CRNS networks) and perhaps provide examples of what would be the minimum number of CRNS probes to be included in a similar set-up in order to assess CRNS SWC spatiotemporal variability in a reliable way at the catchment (in this case 1 km2) scale.*

**AR:** The referee suggests to discuss requirements to the design of dense CRNS network in future experiments. This is an important comment, but the answer strongly depends on the specific aims of such experiments. We have added a paragraph ("Dense CRNS networks are expensive, but feasible in research environments.") to

section 6 in which we discuss very briefly issues of feasibility, also in response to a comment of referee #3 (3.9).

**1.9. Figure 1**

RC: *Figure 1: In the legend you refer to "Climate gauge" and in caption to "Climate station" and in the text (line 169) you use the term meteorological station. Please choose one term only. Besides, from the figure it does not become apparent where are the peaty soils within this catchment are located. From the text we understand that location 23 is on peaty soils, but what is the extend of these? Would it be possible to include the soils on this map too? May be in a separate panel? Finally, the manual samples locations: they seem to be very close to the CRNS sensors, which is ok, but visually it is impossible to distinguish how many you had around each CRNS probe. Could you include information on their number (e.g. in Table 1)?*

AR: We will use the term climate gauge both in the figure, the caption, and the main text. With regard to highlighting the areas of predominantly organic soils, we followed the referee's suggestion to include that information in Tables 1 and 2 (see comment 1.13) which is also in line with suggestions from referee #3 (see comment 3.5).

With regard to the manual sampling locations close to the CRNS sensors, these were always five. Details were explained in Fersch et al. (2020). We added a note to the caption of Fig. 1 to emphasize that fact again.

**1.10. Figure 3**

RC: *Figure 3: Is it correct that the rover can sense neutron intensity between 7000 and 11000 cph? (this is just a question, rather than an improvement suggestion)*

AR: Yes, that was the observational range in the present study context, using the UFZ roving CRNS. We would like to state that this figure as been to moved to the supplementary (now Fig. S4).

**1.11. Figure 5**

RC: *Figure 5: great figure. Could you include the legend also on the right panel for consistency? Also, optionally name the panels as a), b) and c), this will help you shorten the text related to that figure.*

AR: As suggested, we added a legend to the right panel of this figure, and refer to the panels as A, B, and C.

**1.12. Figure 6**

RC: *Figure 6: another very clear figure. However, I wonder why precipitation is presented as mm/6h and the SWC at the panel below in daily values? I guess it is ok if the width of the cell representing a day on the bottom panel is matched with the width 4 bars of rain.*

AR: We think that the vertical bars of 6 hourly rainfall depths provide a helpful supplement to the cumulative precipitation curve, as they display event dynamics more intuitively, while the resolution of 6 hours is a good compromise between hourly and daily. The horizontal width of the bars is matched to the time axis.

**1.13. Table 1**

RC: *Table 1: to address my comment on Figure 1, perhaps you can include an extra column to this table "Dominant soil type" or give percentage of the soil types present within each footprint.*

AR:  Please refer to our response to comment 1.9. We decided to include that information in Tab. 1 (where organic soils appear as Histosols which is the dominant soil type only on the footprint of sensor 23, but is also substantially present in the footprints of sensors 17, 18 and 21).

**1.14. Table 2**

RC:  *Table 2: Under each header add the units, if applicable. That might avoid having a lengthy caption and faster to understand to readers.*

AR:  As suggested, we included the units in the column headers.

**1.15. Other comments**

RC:  *Line 9-11: Do you need to enumerate all the static effects and temporally dynamic factors in the abstract? Shorten sentence if possible.*

AR:  We shortened the sentence so that it reads: "[...] we apply a state-of-the-art procedure to correct the observed neutron count rates for static (heterogeneity in space, e.g. soil organic matter) and dynamic effects (heterogeneity in time, e.g. barometric pressure)."

RC:  *Line 14: you mention already here "constrained interpolation", which is great. Could you also please mention what you will refer later on as "unconstrained" to improve clarity and keep consistent?*

AR:  We agree, though we decided to introduce the term "unconstrained" a bit later in the abstract as it provides a better context to understand the difference. The corresponding sentence is then: "The comparison with independent measurements [...] shows that the constrained interpolation approach outperforms the *'unconstrained'* Ordinary Kriging [...]".

RC:  *Line 44: on "isolated" sensor footprints only. Change "only" for "mainly", as there are already several key studies published using CRNS rovers.*

AR:  We replaced "only" by "mainly".

RC:  *Line 51-57: Am I correct that here you present the main aim of the study? If so, please restructure this paragraph first stating the main aim and then the justification. (the content is good, but the flow can be improved).*

AR:  It is correct that this paragraph outlines the overall motivation and aim of the study. We slightly shortened and restructured the paragraph along the lines suggested by the referee.

RC:  *Lines 67 – 78: The first question or specific objective has been given more space compared to questions 2 and 3. Could you please make this section more concise? In that way the "weight" of the three specific objectives seems equally distributed.*

AR:  We shortened the description of the first objective, but also of the second and third. In summary, we achieved a better balance between the three.

RC:  *Line 94: Abbreviation FDR is not defined previously. Could you please include "Frequency domain reflectometry" here?*

AR:  We included "Frequency Domain Reflectometry" to explain FDR at its first occurrence of in the manuscript.

RC:  *Line 102: change to "We then present and discuss the corresponding results (section 5)..." – as Results*

*and Discussion are presented simultaneously.*

AR: Implemented as suggested.

RC: *Line 109: please provide information on what portion of the catchment is covered with grassland (in percent) and forest (in percent) to give a better idea of the heterogeneity in land use to which you refer in Line 60.*

AR: We added the corresponding percentages of forest (approx. 27 %) and grassland (approx. 69 %) coverage.

RC: *Line 113: Section 3.1. Overview does not add anything to point 3. Data. In fact, you state "as already pointed out". Thus, I would remove the current Section 3.1.*

AR: We removed the former section 3.1 from the manuscript.

RC: *Line 138-142: Section 3.3. is unnecessarily long and can be shortened to two sentences. Alternatively fuse current sections 3.3. and 3.6. as one, giving it an appropriate name. The measurements described in these two sections are used to do the continuous correction of the signal.*

AR: As suggested, we shortened section 3.3 (on incoming neutron flux) and merged it with the former section 3.6 (on meteorological data). Altogether, this became the new section 3.2: "Incoming cosmic-ray neutron flux and meteorological observations".

RC: *Line 144: "As we mention in section 1.1. we require local measurements of the soil water content and the soil bulk density" – you don't actually mention soil bulk density in current section 1.1. Either include it there or rephrase this sentence.*

AR: We thank the referee for pointing out this inconsistency. We prefer to leave the statement in section 1.1 because the transfer function could also be calibrated for gravimetric soil water content, without the need to determine soil bulk density. Instead, we adjusted the above sentence in the section on "Local measurements of soil water content and other soil variables" simply by dropping the reference to section 1.1 (which is not required as the actual calibration procedure is outlined in the methods section). Hence, the above sentence has become: "We require local measurements of soil water content and soil bulk density [...]"

RC: *Line 146: Do "several measurement techniques" refer to the techniques outlines in sections 3.4.1., 3.4.2. and 3.4.3.? If so, please change to "the following measurement techniques".*

AR: As suggested, we changed the sentence to "[...] the following measurement techniques were applied to meet these requirements."

RC: *Line 168: Current section 3.6 should be moved after current section 3.1., as the meteorological data is relevant to all the CRNS analysis. Also, see my comment on using "climatic gauge", "climate station" or "meteorological station".*

AR: Implemented accordingly (merged with section on incoming neutron flux).

RC: *Line 172: Section 3.7 does not really present data (since you do not use it for analysis and merely to create figure 1). I would remove this section as the information is already provided in the caption of Figure 1. Optionally, add the hyperlink from the current section 3.7 to the caption of Fig. 1.*

AR: We agree. Originally, this section was intended to avoid the redundant repetition of the obligatory license acknowledgement. However, we already had to include elongated statements in the figure captions, so we elongated these just a bit more and dropped this section instead.

**RC:** *Line 173: replace "incl." with "including".*

AR:  Was replaced.

**RC:** *Line 411: SMT100 cluster (SoilNet) should be defined/mentioned earlier, as for example in line 90. In line 431 you refer to the same as "FDR-cluster". Choose one term and stick to it.*

AR:  We prefer to stick with FDR, and changed the wording accordingly.

**RC:** *Line 484: "… Fig.5 is both encouraging and disappointing" – put first disappointing and then encouraging, as this is the order in which you present them afterwards. Also, why would you say disappointing about a Figure you include in a publication? Choose a different less negative word. There are several places in the manuscript where you downplay the results of your research (see my comment about line 611 for example).*

AR:  We changed "disappointing" to "unfavourable" and switched the order to "both unfavourable and encouraging".

**RC:** *Line 486: "The vertical shades in the left panel indicate that the theta (Ni) is highly uncertain." You refer here to the percentiles, but which ones/which grey shades? The 5-95 percent or the 25-75 percent ones? Be more specific. Also you can see from this left panel that the uncertainty is larger in the wetter range and smaller in the drier range. Acknowledge that instead of saying that they all are highly uncertain. See my comments about downplaying your results.*

AR:  The discussion focuses on the dark shades (25-75 percentiles) while the light shades are rather for illustrative purposes. We clarified this in the revised version, and also provide further explanation in the caption of Fig. 4 (formerly Fig. 5).

**RC:** *Line 605-607: While limitations are honestly acknowledged the paragraph ends on a negative note, which does not reflect the main outcome. I would recommend rephrasing along the lines "While both the unconstrained and the constrained approaches failed to fully simulate the dry are in the southwest of the SoilNet area, the constrained captured it relatively better."*

AR:  We tend to disagree with that suggestion. This is maybe a misunderstanding, because our sentence "Both models fail to reproduce the dry region in the eastern part of the area" refers to the eastern part, not to the south-western part, as pointed out by the referee. For this eastern dry spot, both interpolation approaches fail the same, so the statement suggested by the referee does not apply. We also do not think that it constitutes a problem that the last sentence of the paragraph addresses an issue that is not a "success". We hence prefer to keep the statement as is.

**RC:** *Line 611: "largely" outperform seems like an overstatement (downplaying the overall good results of your study). I would remove "largely". If you mean that the unconstrained model outperforms during a prolonged period of time, modify text accordingly.*

AR:  We removed the term "largely", as suggested by the referee.

**RC:** *Line 622: typo. Change "reliably" to "reliability".*

AR:  We fixed the typo.

**RC:** *Line 629: I understand the willingness to emphasize the "heterogeneous" in here, as it is a key word in this study. However, not entire convinced about "heterogeneous hydro-meteorological conditions". Are you not considering that precipitation, temperature and barometric pressure measured at the climate station are all the same for all sensors? I would recommend saying "dynamic" or rephrase.*

AR:     The referee is correct that the term "heterogeneous" might in this context be misleading as it implies heterogeneity in space while we mean, as the referee correctly suspected, heterogeneity in time. However, due to the shortening of the paragraph, the sentence is no longer part of the revised manuscript anyway.

RC:     *Line 725: typo. Change "August 2910" to the corresponding year.*

AR:     We fixed the typo.

**2.     Response to referee #2**

**2.1.     Sensitivity factor**

RC:     *Sensitivity factor was assumed to be a constant for each sensor, which seems intuitively reasonable but needs scrutiny. Since these factors are essential to the uniform calibration, it at least requires some citations and/or explanation.*

AR:     The detector-specific sensitivity or efficiency is a result of manufacturer-level variation of detector gas density, geometrical variation, and configuration parameters related to electricity. All of those were fixed once during manufacturing and cannot change over time. [Schrön et al., 2018] have shown that the resulting count rate efficiency is different from sensor to sensor, while significant variation in time is not evident. This is also known from experience with long running sensors of the COSMOS network in periods of more than a decade, where the duration of our 2-month campaign is negligible. On the basis of these explanations, we think that the application of constant, sensor-specific efficiency values is sufficient.

    We added a brief clarification and citation in section 4.1.1 of the manuscript ( Standardization of sensitivity).

**2.2.     Local uncertainty**

RC:     *Please define/specify "local uncertainty". I think the "local" refers to the parameter space, not the spatial-temporal space. If I am correct, it creates some ambiguity in the text since the discussions are always related to space/time in this paper. (Line 271, 415, 665, etc.)*

AR:     We apologize for the ambiguity, and we agree that the term "local uncertainty" requires a clearer definition. In fact, we refer to "local uncertainty" as the uncertainty of our soil moisture estimate for a specific sensor footprint $i$, i.e. as the uncertainty of $\theta(N_i)$ and $\theta_i^{obs}$, expressed as the width of the interval between two quantiles. It should also be clear that this uncertainty only refers to the point in time at which the manual soil moisture measurement campaign took place which was the basis of the $N_0$ calibration.

    In the revised version of the manuscript, we explained the term "local uncertainty" in section 5.1 (formerly 5.2) where the actual results are discussed, with additional details in the caption of Fig. 4 (formerly Fig. 5). In section 4.3, l. 271 of the original manuscript, we dropped the use of the term instead of already defining it. That is because we think that the meaning of the term becomes more tangible for the reader in the context of the presentation of the actual results, hence we do not introduce it before.

**2.3.     Monte-Carlo simulation**

RC:     *Please briefly explain the choice of "200 times" of the Monte-Carlo simulation on the sensitivity of N0. To my understanding, the number of simulations depends on the dimensions of the parameter space. Why are 200 times good enough to quantify the uncertainty of $N_0$ concerning these many parameters and*

*disturbances.*

AR: The referee is correct that it would be good to have a formal justification of the number of runs that constitute our Monte-Carlo analysis. However, the parameters, their assumed probability distributions, and the corresponding stochastic disturbances are very different among each other. That makes it difficult to apply any formal framework to assess the required numbers of runs. For example, some disturbances are rather a sub-sampling (e.g. the determination of the time interval over which the neutron count rates are averaged, or the selection of soil profiles that are included in the interpolation); for other input parameters, the definition of the underlying distribution and its parameters is necessarily arbitrary (e.g. the Kriging range, or the water equivalent from soil organic carbon and lattice water). Given these difficulties, we have addressed the issue rather pragmatically: we found that the results of the Monte-Carlo simulations with using 200 runs are robust, meaning that they do not vary substantially from simulation to simulation with regard to the output we were looking at (which is specifically the interquartile range, while the range between the 5th to the 95th percentile is purely for illustrative purposes). We also found that the results did not substantially change when we increased the number of runs per Monte-Carlo simulation. We have added a brief justification along these lines in section 4.3. Furthermore, in the conclusions of the original manuscript, we had already been open with regard to the arbitrariness of some decision for the design of the Monte-Carlo analysis, and these statements are also part of the revised version of the conclusions.

We would also like to emphasize that the Monte-Carlo-analysis is, in the context of this study, of rather qualitative relevance: its main purpose is to demonstrate that the disagreements that we observe in Fig. 4 can mostly be explained by the local uncertainties of $\theta(N_i)$ and $\theta_i^{obs}$, while location 7 is clearly different.

**2.4. 2.4 Models**

RC: *Line 88 References for the concept of geophysical inversion are needed.*

AR: We now cite [Zhdanov, 2015] in this context as a reference to the fundamental idea of geophysical inversion.

RC: *Line 329 There are three parameters for a variogram model, nugget, sill, and range. The paper only emphasized the range but did not mention the other two. Please specify the parameter selections.*

AR: We apologize for the incomplete documentation. We did not specify nugget and sill, as these do not affect the result of the predicted variable, but only the Kriging variance. Since we do not use the latter, nugget and sill can be chosen arbitrarily (in our case: nugget=0, sill=1). Nugget and sill become important when a theoretical variogram model is fitted to an empirical semi-variogram, as the choice of nugget and sill might affect the range, when the three parameters are fitted together. In our case, we did not fit a variogram model. Instead, the choice of the range of 300 m was rather a preference to express the scale at which we are interested in representing soil moisture heterogeneity.

In the revised version of the manuscript, we state the values of nugget and sill used for our calculations (in sections 4.1.4 and 4.4.1).

RC: *Line 297 The Kriging ranges for soil moisture and bulk density are quite different. Please justify this selection.*

AR: The sampling intervals for the Kriging ranges in the Monte Carlo analysis (section 4.3) were based on the Kriging range values used for the interpolation of the soil variables as outlined in section 4.1.4, which were 50 m for all soil variables except soil moisture (150 m). These values were obtained heuristically from trial-and-error, as already indicated in the original manuscript (ll. 222-228). The arbitrariness of this approach was addressed by defining a sufficiently large interval around these range values from which we would

sample in the Monte-Carlo analysis. We agree, however, that the width of the sampling interval should be set consistently, even if the choice is arbitrary. We decided to reset the intervals to $\pm 50\%$ of the original range values, as outlined in section 4.3 of the revised manuscript which led to minor changes in the results of the Monte-Carlo analyis.

**2.5. Footprint, model parameters, and scaling**

**RC:** *One of the unique features of CRNS is its large footprint, which could directly influence data visualization, model selection, and interpolation. The grid size for the interpolation process is 10 m \* 10 m (Line 311), which is much smaller than the footprint. This implies that the modeling is not just an interpolation but also involves a downscaling process for the CRNS measurements. It is of great interest in terms of the CRNS studies. However, it also requires more clarification and cautiousness. For example, is it reasonable to use observed soil moisture, $\theta(N_{obs})$, to do Ordinary Kriging with a resolution much smaller than its footprint? Does it implicitly assume that observed soil moisture values are also representative at a smaller scale?*

**AR:** We thank the referee for this comment. Obviously, he or she is entirely right to demand that cautiousness. We had hoped to express that caution with our statement from ll. 311-314 of the original manuscript:

> [ll. 311-314] The grid resolution is arbitrarily selected, and does not necessarily reflect the resolution at which the grid effectively conveys information of spatial heterogeneity; in other words, the product should not be interpreted at the scale of 10 m. Still, we require this comparatively fine horizontal resolution since some of the following steps require to re-aggregate (i.e. to average) the spatial soil moisture estimates inside a CRNS footprint.

This paragraph is still part of the manuscript and we hope that it sufficiently addresses the referee's comment. We do not actually aim to represent soil moisture variation between 10 m grid tiles, but we require that resolution in order to reasonably apply the forward operator to obtain neutron intensity from a spatial soil moisture grid. In addition, one could see this sub-footprint resolution as a tool to represent gradients in the footprint, rather than values of the single cells.

Furthermore, we emphasized in ll. 328-332 (original manuscript) that Kriging is used as a "model" to represent our notion how soil moisture varies at a specific scale:

> [ll. 328-332] In this study, let us assume that the spatial distribution of soil moisture in the study area is smooth and continuous, and that this spatial pattern could be represented by a model $m$ that corresponds to Ordinary Kriging with an exponential variogram model and a range parameter of, say, 300 m, using the CRNS sensor locations as points of support. We hope it is clear to the reader that the choice of such a model is arbitrary and subjective, although it should be based on our "expert" notion of how soil moisture varies at a specific scale.

Finally, the referee wonders whether we "implicitly assume that $[\theta(N)$ is] representative at a smaller scale". Our answer would be no, although it is true that the unconstrained model, technically, reproduces $\theta(N_i)$ at the sensor location $i$. However, that is rather a side effect and not a necessary requirement. The key property of our model $m$ is that it represents soil moisture variation at a scale that is given by our (arbitrary) choice of the variogram (exponential with a range of 300 m).

**RC:** *The design of the forward operator and the optimization argument is innovative since it provides a way of downscaling CRNS measurement to almost any arbitrary scale/resolution, which may be only limited by computational capacity.*

*The design of the dense network made the footprints of CRNS largely overlapped, which provides extra information about soil moisture spatial patterns. This may also make it logically possible and reasonable to do the downscaling and to improve the interpolation. Can the overlaps be used for results validation?*

**AR:** We agree, in general, with the referee's view that the use of the forward operator allows for a certain level of downscaling (see our response to comment 2.5), and it certainly is one of the specific aims of this study to demonstrate that potential. However, we do not think that the achievable resolution is purely a matter of computational resources. In our view, it is rather a matter of how well our model $m$ is able to represent patterns at high resolution (see also our response to the previous comment). Example: We could enhance, in our setup, the spatial resolution of our target grid from $10\,\mathrm{m}$ to let's say $10\,\mathrm{cm}$. That would involve a substantial increase in computational costs for the interpolation and the application of the forward operator, yet the effective/meaningful resolution of the results will not be any higher than before.

Somewhat related to that point is the aspect of overlap: in general, we would expect that a large overlap from multiple sensors would help to better constrain the inverse problem, yet it does not, in our view, provide "extra information" for an independent validation. Even with a strong overlap, we still need a model of spatial soil moisture variation to make the problem solvable. The advantage of the overlap is particularly that the parameters of that model will probably be constrained better because changes of soil moisture in the region of overlap will affect multiple footprints and hence multiple values of $N^{sim}$.

While this discussion is certainly interesting, we would prefer not to extend it further in the context of the manuscript. We see the present study as a proof-of-concept, and both practical and theoretical aspects should be explored in future studies, as also outlined in ll. 722-730 of the original manuscript and in the conclusions section of the revised manuscript.

**2.6. Technical Comments**

**RC:** *Line 26 "small spatial measurement support" and Line 330 "points of support". Support is an important concept in defining spatial scales of soil sampling and measurements. I recommend adding a definition and citations here. This would also help to present the results on soil moisture spatial patterns in the following sections.*

**AR:** In the revised manuscript, we refer to [Blöschl and Grayson, 2000] in section 1.1., as the key reference with regard to the concept of spatial support in the observation of spatial variables. We have replaced the term "points of support" by "data points" or "data points for the interpolation" throughout the manuscript, also in line with terminology used by [Blöschl and Grayson, 2000].

**RC:** *Line 259 "assuming a spatially uniform value of $N_0$..." Modification required. Since $N_0$ mainly depends on the sensor itself after correcting all factors (air pressure, vegetation, lattice water, etc.), it is not a spatial variable.*

**AR:** We agree that this is misleading. We dropped "spatially" so the statement becomes "[...] assuming a uniform value of $N_0$ [...]".

**RC:** *Line 262 Eq. 1 - recommendation: replace comma with semicolon, i.e. $\theta(N_i; N_0)$. To my understanding, $N_i$ is a variable, and $N_0$ is a parameter in Eq. 1.*

AR: We thank the referee for the suggestion, but we would prefer to keep the notation as it is: while in the context of Eq. 2, $N_0$ is the parameter (which is optimized), the constrained interpolation treats all $N_i$ as parameters.

RC: *Line 350 delete extra "suitable"*

AR: Thanks, was deleted.

RC: *Line 622 reliably -> reliability*

AR: Was corrected.

**3. Response to referee #3**

**3.1. Advancement of practical applications**

RC: *This manuscript describes the first ever attempt to use multiple CRNS to estimate catchment-scale soil moisture, including temporal and spatial variations. The research conducted is of interest and importance to the research community, even if the findings are somewhat marginal in their advancement of practical application.*

AR: The campaign itself and the present analysis were designed to answer primarily theoretical and methodological questions of cosmic-ray neutron sensing. Hence, the study was not geared towards practical applications. Since the reviewer questions the advancement made by this study, we would like to take the opportunity to explain the highlights and the potential for continued future research.

We agree that it will, for the foreseeable future, not be cost-effective or practical to deploy 10-20 CRNS probes per square kilometer for routine soil moisture monitoring (see also our response to comment 3.9). And yes, the results of the present analysis, in terms of the soil moisture patterns retrieved from our CRNS network, might not appear overwhelming, as opposed to the spatial detail that could be achieved with wireless sensor networks (see also our response to comment 3.8).

As the referee pointed out correctly, this analysis is the first attempt to use the observations of such a dense network for a spatially explicit retrieval of soil moisture patterns, and it has a clear methodological focus:

1. we demonstrate a uniform homogenisation and calibration approach across multiple CRNS stations (a concept that will be relevant particularly for future CRNS observatory networks), and

2. on that basis, the potential to capture main features of spatial soil moisture variability at the scale of hectometers, continuous in time.

3. We also introduce an entirely new approach to constrain a spatial interpolation by the dynamic CRNS footprint characteristics.

In contrast, we do not claim to provide a scalable monitoring technique; we do not claim to represent small-scale features of soil moisture variability (as clearly stated in section 4.4: the interpolation output should not be interpreted at the technical resolution of $10\,\text{m} \times 10\,\text{m}$ (also cf. to sect. 2.5 of this letter); and we do not claim to consistently link our observations to hydrological and soil hydraulic processes (although we briefly discuss potential explanations of temporal dynamics in section 5.3 of the original manuscript, now section 5.2).

While this study does explicitly not claim to *achieve* these – certainly desirable – goals, it is still *motivated* by them and works towards those goals. For example, future research should aim to combine such CRNS observations with spaceborne remote sensing data: to obtain patterns at a higher resolution via downscaling, or to upscale CRNS observations beyond such local networks in a cost-efficient way. And we should design assimilation experiments with hydrological models at various levels of physical detail in order to understand how we can consistently represent our observations and our physical expectations. We do not solve these issues in the present study, but we outline a methodological framework that should facilitate them.

We have tried to revise the conclusions section in a way that these implications for prospective research become clearer.

**3.2. Length of the manuscript**

RC: *While the paper is well-organized and written, it is excessively long and tedious to read. I understand that the authors wish to share the minutia of their novel methodology, but as written the paper is difficult to read and seems more akin to a grant proposal than a scientific manuscript.*

AR: We thank the referee for this comment. It is in line with the comments of referee #1 who also suggested to shorten the manuscript. Then again, referee #2 rather required *more* details in terms of both methodology and discussion. Still, we agree there was ample potential to make the paper more concise, and have revised it accordingly. At the same time, we tend to insist that due to the strong methodological focus of the paper, the adequate documentation of methodological details and the discussion of their limitations should not suffer. We hope that we have found a balance that meets the requirements of the referee.

**3.3. Absolute values**

RC: *I dislike the goal of the project to match the pattern rather than absolute values (stated in Line 434) and the general avoidance of quantifying differences or variability in the estimated and measured soil moisture values. I understand why the authors have chosen this approach, but I also think that the use of the absolute values of measured and estimated volumetric water content would be useful to present to the scientific community. Relative values only provide so much information.*

AR: There is maybe a misconception of the study, as it has not been "the goal of the project to match the pattern rather than absolute values", and from our perspective there is no "general avoidance of quantifying differences or variability" in the manuscript.

The reviewer refers to line 434 of our original manuscript:

> [...] to eliminate the potential effect of systematic bias in the SoilNet data, [...] we target the matching of the pattern rather than the absolute values."

This statement only applies to the comparison to the SoilNet data, and explains why we use Spearman's rank correlation coefficient as an objective measure of similarity between the soil moisture inferred from the SoilNet and the interpolation of CRNS-based soil moisture estimates: in this case, we are specifically interested in how well the spatial patterns agree, in order to evaluate which interpolation method better represents soil moisture gradients at the hectometer scale. Numerous equations exist for converting permittivity (the prime observational variable of the SoilNet) to volumetric soil moisture [Mohamed and Paleologos, 2018], and the exact shape could differ substantially, depending on underlying functions and corresponding coefficients. To eliminate the associated uncertainty and arbitrariness from our analysis, we chose to use Spearman rank

correlation.

But we agree that the above formulation is ambiguous, so we changed the paragraph to:

> We evaluate the similarity of the spatial soil moisture patterns obtained from the FDR-cluster and the interpolation of $\theta(N_i)$ for each day from May 20 to July 15, 2019. As a measure of similarity, we chose Spearman's rank correlation of the corresponding soil moisture grids. Using that measure, we eliminate potential effects of uncertainty in the soil moisture values obtained from the SoilNet, as the conversion from permittivity to volumetric soil moisture can be subject to systematic bias [Mohamed and Paleologos, 2018].

While we are convinced that this comparison approach is adequate in the context of this study, we would still like to address the referee's concern of a "general avoidance of quantifying differences or variability in the estimated and measured soil moisture values." To that end, we would like to adopt the referee's suggestion made in comment 3.8: *"[...] I would be more interested in a time series figure showing the mean field-scale volumetric water content for each of the three scenarios- unconstrained, constrained, and SoilNet. I suspect that the constrained and unconstrained values would be far more similar to one another than to the SoilNet values."* Certainly, the referee is correct with assuming that the two CRNS-based interpolation results (constrained, unconstrained) are far more similar to each other than each of them is to the SoilNet estimates. We think that this notion is obvious from the soil moisture maps shown in Fig. 8 (formerly Fig. 9).

Nevertheless, we would like to meet the referee's demand and added another panel to Fig. 9 (formerly Fig. 10) in which we show the soil moisture time series for the three scenarios for the area of the SoilNet. We interpret the term "mean field-scale volumetric water content", as suggested by the referee, as the mean over the SoilNet area.

**3.4. Shallow groundwater**

**RC:** *Section 2: The authors mention in passing the presence of very shallow groundwater, but do not mention how this likely has significant effects on their CRNS measurements. This issue should be addressed much more thoroughly.*

AR: We agree that the interpretation of soil moisture patterns would benefit from a better knowledge of the depth to the groundwater table. Unfortunately, we rely on the references given in section 2 of the original manuscript. Spatially explicit information on depth to the groundwater table are not available, and we assume that the below-ground structure is complex and heterogeneous: the very shallow groundwater, where present, appears to result from the local accumulation of percolating water on low-permeable soil layers – a perched aquifer, if at all. As such, it would not be a laterally extended water body that feeds water to the top-soil but rather a part of the soil water balance. All this information, however, is patchy, and rather hypothetical, and does not rely on robust spatial observations of shallow groundwater. Hence, we would prefer not to excessively hypothesize about this matter in the manuscript. However, we added a statement in section 2 on the possibility of such local structures, and the lack of corresponding data.

**3.5. Soil texture and SOM**

**RC:** *Additionally, the authors should provide the textural and SOM information for samples taken near each CRNS (texture) and for each mixed sample (SOM). Providing an average catchment-scale value is not acceptable, and readers cannot be expected to read every cited paper to find this information, which could*

*easily be incorporated into Table 2.*

AR: We agree that it would be helpful to explicitly link soil attributes to CRNS locations in order to allow for a better interpretability. A soil texture analysis was unfortunately not carried out for the soil that was sampled by cylinders near each CRNS. Using the horizontal weighting scheme from [Schrön et al., 2017] we have, however, extracted the dominant soil types from the Übersichtsbodenkarte (soil survey) (1:25,000) of the Bayrisches Landesamt für Umwelt [Bayrisches Landesamt für Umwelt, 2014] and the dominant soil texture classes from the Bayrisches Landesamt für Digitalisierung, Breitband und Vermessung within the product Bodenschätzungsdaten [Landesamt für Digitalisierung, Breitband und Vermessung, 2018]. The soil organic matter (SOM) content was only determined for mixed samples from the three landuse/soil classes (forest/mineral, grassland/mineral, and grassland/organic).

In order to meet the referee's demand, we included the SOM content (weighted average in the footprint) as an additional attribute in Tab. 2. As for the soil texture class and type, we included the dominant soil texture class and soil type in Tab. 1 (instead of Tab. 2), as the dominant land use type in the footprint is also provided in that table.

RC: *Further, the estimated gravimetric water content values > 1.0 g/g shown in Figure 4b are unrealistic, unless you are considering a highly organic soil. However, based on the information presented, it is impossible to determine the soil type near CRNS 21 and 23.*

AR: The referee is correct in noting that the high gravimetric soil water content values for the CRNS sensors 21 and 23 are due to the fact that the footprints of these sensors are dominated by highly organic soils. With the additions to Tab. 2 as suggested above, this should become more transparent to the reader – so we thank the referee for this suggestion.

RC: *Table 2: In addition to including information regarding soil texture and SOM for each site, the authors should include some indication of the variability of the values shown for each site.*

AR: Quantifying the variability of soil texture and SOM in the sensor footprint is difficult due to the limited amount of data (as pointed out above): high-resolution soil mapping was, unfortunately, not part of the underlying campaign. Hence, we suggest to stick with the additional information in Tab. 2 as outlined in our response above.

RC: *Also, bulk density values are incredibly low, less than 1 kg/L in most cases. Including the SOM content in this table would make those values look less suspect, if indeed the SOM content is extremely high. If not, the authors need to address the very low bulk density values reported.*

AR: The bulk density values shown in Tab. 2 are in fact rather low. It is important to consider, though, that these bulk density values represent the weighted average bulk density per sensor footprint, with the weights corresponding to the vertical and horizontal sensitivity as given by [Schrön et al., 2017]. The low bulk density values here are hence, partly, a result of the vertical weighting: the bulk density generally *decreases* towards the surface (as can be seen in the vertical profiles shown on the supplementary, Fig. S2), while sensor sensitivity pattern *increases* towards the surface. So while the lowest bulk density values are in fact due to organic, high porosity soils (sensors 21 and 23), the generally low bulk density values in Tab. 2 are also due to the fact that the vertical weighting emphasizes the less-dense top soil. We thank the referee for pointing out this issue. We added a corresponding statement directly in the caption of Tab. 2 in order to clarify this issue.

**3.6. Uncertainty of $\theta^{obs}$**

**RC:** *Figure 5: How does the variability in $\theta_i^{obs}$ during Monte Carlo simulations compare to the variability observed in measured volumetric water content from thermo-gravimetric samples?*

**AR:** $\theta_i^{obs}$ or its variability in the Monte Carlo analysis should not be compared to the volumetric soil moisture content as obtained from the thermo-gravimetric samples. As pointed out in section 3.4.2 of the original manuscript, we only have *one* thermo-gravimetric sample close to each CRNS sensor. $\theta_i^{obs}$, in turn, is the result of an interpolation of thermo-gravimetric samples *and* FDR-measurements, and subsequent vertically and horizontally weighted averaging. Hence, we do not see the reason for such a comparison.

**RC:** *Could the authors provide mean uncertainty values for Monte Carlo $\theta(N_i)$ and $\theta_i^{obs}$? It is difficult to estimate these values from Figure 5 alone.*

**AR:** We assume that with *"mean uncertainty values for Monte Carlo $\theta(N_i)$ and $\theta_i^{obs}$"*, the referee refers to the mean values of the inter-quartile range for the realizations of $\theta(N_i)$ and $\theta_i^{obs}$ in the Monte-Carlo analysis (also see our response to the comment 2.2 of referee #2 in which we clarified what we mean by local uncertainty). We have added these mean widths of the inter-quartile ranges (0.08 m³/m³ for $\theta(N_i)$, 0.03 m³/m³ for $\theta_i^{obs}$) to the main text in section 5.1. We would like to maintain, though, that the main motivation of Fig. 4 (formerly Fig. 5) is to visually demonstrate that the disagreements that we observe in Fig. 3 (formerly Fig. 4) can mostly be explained by the local uncertainties of $\theta(N_i)$ and $\theta_i^{obs}$, while location 7 is clearly different. Average uncertainty values are hence, in our opinion, only of limited value in this context.

**RC:** *Line 543: It would be good, again, if the authors would quantify this uncertainty.*

**AR:** As we have quantified the average uncertainty of $\theta(N)$ values in response to the referee's previous comment (implemented in section 5.1 of the revised manuscript), we prefer not to repeat the value again on this occasion. We hope that the referee will agree.

**3.7. Reference evapotranspiration**

**RC:** *Line 525 and Figure 6a: Reference evapotranspiration is different than potential evapotranspiration, but the terms are used interchangeably in the text. Make sure all instances are changed to "reference."*

**AR:** We changed all instances to "reference evapotranspiration".

**3.8. Comparison to SoilNet**

**RC:** *Figure 9. The results shown in this figure are underwhelming. I expected that the highly concentrated CRNS sensors would be able to provide a closer match in spatial soil moisture distribution to the Soil-Net measurements. The lack of spatial agreement with SoilNet soil moisture values, even with a high concentration of CRNS that is unlikely to replicated in practice, is surprising and a bit disappointing. If the CRNS are unable to provide useful spatially explicit information, what is the benefit of using these extremely expensive sensors rather than many cheaper in-situ sensors or downscaled remote sensing data? Also, I would be more interested in a time series figure showing the mean field-scale volumetric water content for each of the three scenarios- unconstrained, constrained, and SoilNet. I suspect that the constrained and unconstrained values would be far more similar to one another than to the SoilNet values.*

**AR:** We have already responded to parts of this comment in our response to comments 3.1 and 3.3, so we would like to keep this response brief. We understand that the visual impression from Fig. 9 might be disappointing

at first - but only if you expected to reproduce the meter-scale variability as represented by 300 invasive sensors (SoilNet, 50 profiles, 3 depths, 2 redundant measurements per depth) by 4 (inside) or 6 (near SoilNet area) non-invasive hectare-scale CRNS sensors. We hope that the referee does not misinterpret our goal as "advertising" dense CRNS networks as "the" solution (according to the comments of referee #1, we are excessively critical to our results rather than sugarcoating them).

As matching meter-scale variability cannot be the goal of a study using just a few hectometer-scale CRNS stations, we would like to clarify again: Our main finding is that dense CRNS networks can help to represent soil moisture gradients at the hectometer scale, in an area of say $1 \, km^2$, and that the inversion-style technique of constrained interpolation can add to that. In combination with auxiliary information, such as remote sensing, we might even be able to add further details to such patterns – corresponding studies are underway.

Please allow a short remark regarding the mentioned proportionality, practicability, and costs: a high number of in-situ sensors such as the WSN/SoilNet can also become very expensive in terms of long-term maintenance and disruptive in terms of installation, which are clear limitations especially for intensely managed agricultural fields. In contrast, CRNS sensors require hardly any maintenance and sense soil moisture non-invasively at a larger scale, while they can be used in small networks or mobile roving campaigns. We think that these advantages could be well suited for agricultural applications as well as evaluation of remote-sensing and modeling products, where intermediate-scale soil moisture patterns are of key importance.

**3.9. Costs**

**RC:** *Line 595: While I agree that the current application of a large number of CRNS in a small area is interesting, I do not think it is economically feasible or sustainable. The authors should at least mention the cost prohibitive nature of this study.*

AR: As already pointed out in our response to comment 3.1, we agree that such dense networks are not suited for routine soil moisture monitoring, and while we have not suggested otherwise in our manuscript, we explicitly clarify that in the revised version. To that end, we added a dedicated paragraph in the conclusions section, in which we outline this limitation, but also the potential role of dense networks in research contexts where they could contribute to a better understanding of soil moisture variability in space and time.

**3.10. Technical corrections**

**RC:** *Line 301: Should read "how the availability of sampling locations affected N0 calibrations [...]*

AR: Thanks for pointing out this incomplete sentence, we changed it accordingly.

**RC:** *Line 725: Should read "from August 2019"*

AR: The typo was corrected, thanks for pointing it out.

**4. Response to the Editor**

First of all, we would like to thank the Editor not only for managing the review process, but also for taking the time to also provide specific comments with regard to the manuscript. Please find our responses to each comment below.

**EC:** *[...] Most of the criticisms refer to the fact that overall the text is too long (with respect to the core*

*information provided, I would add) and also with a few long-winded parts. This is something that has to be necessarily fixed in the revised version (as you also mentioned in your preliminary responses).*

AR:   We have followed the specific and constructive suggestions of referee #1 to shorten the manuscript, and have furthermore attempted to make the paper more concise while keeping the required detail. Then again, some requests, specifically by referee #3 and the Editor, required to add additional information in tables and figures that required also additional text. In total, however, the manuscript became substantially shorter (by five pages).

EC:   *All of the reviewers judged very positively the scientific significance and quality of your study, but the presentation of your results requires refinement, and I suggest should be even more effective. When presenting your methods and the relevant outcomes, you should try to follow a more precise sequence of arguments.*

AR:   We have attempted to consider this comment throughout the manuscript: by restructuring the methods sections, by better aligning the structure of the sections "Introduction" and "Results and discussion", following the suggestions made by referee #1 (see 1.6), and by entirely reshaping the conclusions section.

EC:   *Moreover, I think that comparisons between SoilNet and CRNP datasets should be discussed more in-depth. Would a power spectrum help ascertain and better understand possible agreements between the fluctuations of the SoilNet point data with respect to those of the CRNP field values?*

AR:   We would like to thank the Editor for the suggestion. It might in fact be possible that more advanced techniques of time series analysis could help to better understand agreements and disagreements in the dynamics between the soil moisture series obtained from the SoilNet and the CRNS interpolation. In our view, however, the comparison to the SoilNet constitutes only one, and not the most important, component of this study. The most important feature is to demonstrate the spatio-temporal retrieval of soil moisture from a dense CRNS cluster, including a consistent data homogenization approach. We would hence like to refrain from expanding the scientific and methodological scope of the present manuscript which is already, as raised by the Editor and two referees, very extensive. We sincerely hope that follow-up studies will be able to focus more on this issue. Specifically, we implemented a similar experimental design in the Wüstebach catchment, Western Germany, where a cluster of 16 CRNS is combined with an extensive SoilNet cluster of 150 nodes.

EC:   *If I understood well, you do not have the physical properties of soils in your study area, although this is a task that can be quite easily performed, and at least the textural information can be measured with low experimental burdens.*

AR:   While the determination of soil texture as such might be scientifically undemanding, a systematic mapping of the study area is certainly resource-intensive, and not feasible for us in our current project and study context. And while we agree that a detailed soil texture map could offer interesting opportunities to better understand small scale variability of soil moisture, we would also like to maintain that such an analysis is beyond the scope of the present study. In order to address this comment, however, we would like to refer to our response to comment 3.5 of referee #3 in which we outline how we incorporated additional soil information in tables 1 and 2 of the manuscript.

EC:   *Can your SoilNet data together with some covariates help identify somehow better the penetration depth of the CRNP for your study area and, more importantly, how the CRNP penetration depth changes as soil moisture values change?*

AR:   We would like to emphasize again that we would prefer not to expand the focus and scope of the study much further towards the analysis of the SoilNet data. Better understanding the penetration depth and hence the

vertical representativeness of CRNS-based soil moisture estimates is an important line of research, yet it is not at the heart of this study. So while we appreciate the good ideas suggested by the Editor, we feel that these additional analyses will make it difficult to keep the focus of the manuscript (which is already challenging in its present form).

Having said this, we attempted to consider the Editor's suggestion by including, in Fig. 9 of the revised manuscript (formerly Fig. 10), a time series of the range of CRNS penetration depths across the SoilNet area, as computed for the footprints of those six CRNS sensors which are closest to the SoilNet area and hence affect the result the most. The different penetration depths are, here, mainly a function of different bulk densities in the sensor footprints and of course of the changes in soil moisture over the study period. We combined this additional analysis with the request of referee #3 to also show the temporal dynamics of average soil moisture in the SoilNet area for both the SoilNet observations and the CRNS-based observations. More details are provided in section 5.3.2 of the revised manuscript.

**References**

[Bayerisches Landesamt für Umwelt, 2014] Bayerisches Landesamt für Umwelt (2014). Übersichtsbodenkarte tk25-blatt 8132. www.lfu.bayern.de.

[Blöschl and Grayson, 2000] Blöschl, G. and Grayson, R. (2000). Spatial observations and interpolation. In Blöschl, G. and Grayson, R., editors, *Spatial Patterns in Catchment Hydrology - Observations and Modelling*, chapter 2, pages 17–50. Cambridge University Press, Cambridge.

[Fersch et al., 2020] Fersch, B., Francke, T., Heistermann, M., Schrön, M., Döpper, V., Jakobi, J., Baroni, G., Blume, T., Bogena, H., Budach, C., Gränzig, T., Förster, M., Güntner, A., Hendricks-Franssen, H.-J., Kasner, M., Köhli, M., Kleinschmit, B., Kunstmann, H., Patil, A., Rasche, D., Scheiffele, L., Schmidt, U., Szulc-Seyfried, S., Weimar, J., Zacharias, S., Zreda, M., Heber, B., Kiese, R., Mares, V., Mollenhauer, H., Völksch, I., and Oswald, S. (2020). A dense network of cosmic-ray neutron sensors for soil moisture observation in a pre-alpine headwater catchment in germany. *Earth System Science Data*, 12:2289–2309.

[Landesamt für Digitalisierung, Breitband und Vermessung, 2018] Landesamt für Digitalisierung, Breitband und Vermessung (2018). Bodenschätzung. https://geoservices.bayern.de/wms/v1/ogc_alkis_bosch.cgi?

[Mohamed and Paleologos, 2018] Mohamed, A.-M. O. and Paleologos, E. K. (2018). Chapter 16 - dielectric permittivity and moisture content. In Mohamed, A.-M. O. and Paleologos, E. K., editors, *Fundamentals of Geoenvironmental Engineering*, pages 581–637. Butterworth-Heinemann.

[Schrön et al., 2017] Schrön, M., Köhli, M., Scheiffele, L., Iwema, J., Bogena, H. R., Lv, L., Martini, E., Baroni, G., Rosolem, R., Weimar, J., Mai, J., Cuntz, M., Rebmann, C., Oswald, S. E., Dietrich, P., Schmidt, U., and Zacharias, S. (2017). Improving calibration and validation of cosmic-ray neutron sensors in the light of spatial sensitivity. *Hydrology and Earth System Sciences*, 21(10):5009–5030.

[Schrön et al., 2018] Schrön, M., Zacharias, S., Womack, G., Köhli, M., Desilets, D., Oswald, S. E., Bumberger, J., Mollenhauer, H., Kögler, S., Remmler, P., Kasner, M., Denk, A., and Dietrich, P. (2018). Intercomparison of cosmic-ray neutron sensors and water balance monitoring in an urban environment. *Geoscientific Instrumentation, Methods and Data Systems*, 7(1):83–99.

[Zhdanov, 2015]  Zhdanov, M. S. (2015). Chapter 1 - forward and inverse problems in science and engineering. In Zhdanov, M. S., editor, *Inverse Theory and Applications in Geophysics (Second Edition)*, pages 3–31. Elsevier, Oxford, second edition edition.

---

## Referee Report (RR1)

Second round of review to Heistermann et al., 2021 "Spatio-temporal soil moisture retrieval at the catchment-scale using a dense network of cosmic-ray neutron sensors"

As pointed out in my first review of this paper the paper is of

1) Scientific significance – excellent
2) Scientific quality – excellent

And after the implemented changes

3) Presentation quality – very good/excellent

The authors have addressed all my comments and the manuscript reads much better. Summary and conclusions reads very well and is much improved.

There are only two very small remarks and I would say addressing them is optional.

Line 43: Small typo "(Schrön et al., 2018, see, e.g.,).", change to (see e.g. Schrön et al., 2018)

Line 517: "section S2" add "section S2 in supplementary", as you have done throughout the text.

I have no further comments and I think this article should be published without further revisions.

---

## Author Response (AR2)

**Author reponse to referee comments**

**Spatio-temporal soil moisture retrieval at the catchment-scale using a dense network of cosmic-ray neutron sensors**

Maik Heistermann et al.

*Hydrol. Earth Syst. Sc. Discuss.,* `doi:10.5194/hess-2021-25`
* * *
**RC/EC:** *Reviewer/Editor Comment*,     AR: *Author Response*,     ☐ Manuscript text

Dear Referees, dear Editor,

thank you for the final evaluation and acceptance of our manuscript, and thanks again for all the major and minor contributions and comments along the way.

The following minor changes were made to the manuscript based on the suggestions of referee 1:

- Line 43: We changed "(Schrön et al., 2018, see, e.g.,)." to "(see e.g. Schrön et al., 2018)"

- Line 517: We changed "section S2" to "section S2 in supplementary".

Kind regards,
Maik Heistermann
(on behalf of the author team)